# Cross-national disparities in healthcare workers' perceptions: Examining fear of infection and confidence in the received COVID-19 vaccines amid emerging variants

Mai Hussein[1,2], Assem Gebreal[3], Ahmed Naeem[4], Asmaa Mohammed AboElela[5], Hoda Ali Ahmed Shiba[5], Jargaltulga Ulziijargal [6], Aesha L. E. Enairat [7], Ibrahim Adel[8], Bayan Ayash[9], Shehata Farag Shehata[10], Safar Abadi Alsaleem[10], Ahmed A. Mahfouz[10], Omar Alwakaa[11], Logina Ezz Elarab[12], Vanessa Pamela Salolin Vargas[13], Fabio Massimo Oddi [14], Hala Bakro[15], Dennis Brempong[16], Muhereza Morgan Meike [17], Ramy Mohamed Ghazy [10,18]*

1 Clinical Research Administration, Alexandria Health Affairs Directorate, Alexandria, Egypt, 2 Ministry of Health and Population, Cairo, Egypt, 3 Faculty of Medicine, Alexandria University, Alexandria, Egypt, 4 Al-Azhar Faculty of Medicine, Asyut, Egypt, 5 Public Health and Community Medicine, Community and Occupational Medicine Department, Faculty of Medicine (Girls), Al-Azhar University, Cairo, Egypt, 6 Mongolian National University of Medical Sciences, Ulaanbaatar, Mongolia, 7 Al-Quds University, Jerusalem, Palestine, 8 Faculty of Medicine, Al Neelian University, Khartoum, Sudan, 9 Gulf Medical University, Ajman, United Arab Emirates, 10 Family and Community Medicine Department, College of Medicine, King Khalid University, Abha, Saudi Arabia, 11 Philipps University of Marburg, Marburg, Germany, 12 High Institution of Public Health, Alexandria University, Alexandria, Egypt, 13 Universidad WestHill, Ciudad de México, Mexico, 14 Tor Vergata University of Rome, Rome, Italy, 15 Faculty of Medicine, University of Aleppo, Aleppo, Syria, 16 University of Cape Coast, Cape Coast, Ghana, 17 Mbarara University of Science and Technology, Mbarara, Uganda, 18 Tropical Health Department, High Institute of Public Health, Alexandria University, Alexandria, Egypt

* ramy_ghazy@alexu.edu.eg

## Abstract

Variants of severe acute respiratory syndrome coronavirus 2 (SARS-CoV-2) can influence transmissibility, virulence, vaccine efficacy, the effectiveness of therapeutic agents, diagnostic accuracy, and the overall success of public health interventions. This study aimed to assess the impact of emerging variants on healthcare workers' (HCWs) fear related to SARS-CoV-2 new variant infection and to evaluate their confidence in the received vaccines. A globally distributed cross-sectional study was performed using an online anonymous survey and face-to-face interviews between 1st November and 5th December 2023. The fear level was assessed by the Fear of Coronavirus Disease 19 (COVID-19) Scale (FCS), and the confidence level in the received COVID-19 vaccines was measured using the Arabic Tool for Assessment of Post-vaccination Confidence in COVID-19 vaccines (ARAB-VAX-CONF). A total of 5843 eligible HCWs completed the survey with a mean age of 32.1 ± 10.8 years. Of them, 42.5% were from the Eastern Mediterranean region, 24.2% were from the African region, 14.4% were from the Western region, and 18.9% were from other regions (Eastern Asia and Latin America). Nearly three-fourths (72.7%) were vaccinated,

**Data availability statement:** All relevant data are within the paper and its Supporting information files.

**Funding:** The author(s) received no specific funding for this work.

**Competing interests:** The authors have declared that no competing interests exist.

primarily with Pfizer (40.0%), AstraZeneca (36.8%), and Sinopharm (14.3%). Nearly two-fifths (40.5%) were in extreme fear of catching infection from the COVID-19 emerging variants. Among the HCWs who received COVID-19 vaccines, 41.0% showed good confidence in the received vaccine. Predictors of lower fear included being married [adjusted odds ratio (AOR): 0.8; 95% CI (0.7–0.9)], having a small family of two members [AOR: 0.63; 95% CI (0.5–0.78)] or three members [AOR: 0.62; 95% CI (0.51–0.72)], and being a pharmacist [AOR: 0.75; 95% CI (0.55–0.92)]. Conversely, predictors of increased fear included being divorced or widowed [AOR: 1.3; 95% CI (1.0–1.8)], residing in rural areas [AOR: 1.6; 95% CI (1.4–1.8)] or desert/mountain areas [AOR: 2.5; 95% CI (1.6–4.0)], having insufficient income and in debt [AOR: 2.5; 95% CI (2.2–3.1)], having insufficient income [AOR: 2.4; 95% CI (2.0–2.8)], and having chronic diseases [AOR: 1.2; 95% CI (1.1–1.4)]. Predictors of good confidence in the received vaccine were middle age (30–39 years) [AOR: 1.4; 95% CI (1.1–1.8)], age group 40 years and more [AOR: 1.8; 95% CI (1.4–2.3), rural/other residence [AOR: 1.3; 95% CI (1.1–1.5)], male sex [AOR: 1.3; 95% CI (1.1–1.4)], and small family members of one [AOR: 5.5; 95% CI (4.2–7.2)], two [AOR: 1.5; 95% CI (1.2–1.9)], and three [AOR: 1.3; 95% CI (1.1–1.6)]. On the other hand, having chronic diseases [AOR: 0.82; 95% CI (0.71–0.95)], having mental disorders [AOR: 0.59; 95% CI (0.51–0.69)], high family number of four [AOR: 0.78; 95% CI (0.69–0.89)], personal history of COVID-19 infection [AOR: 0.61; 95% CI (0.53–0.71)], and experiencing side effects of vaccination [AOR: 0.63; 95% CI (0.55–0.72)] were associated with low confidence regarding the received vaccine. In conclusion, HCWs exhibited notable fear of infection with SARS-CoV-2 new variants, along with low confidence in the vaccine. The study suggests realistic approaches, such as targeted interventional programs to address the fear, resolve uncertainties, and promote widespread vaccine confidence among HCWs.

## Introduction

Coronavirus Disease 2019 (COVID-19) is still present worldwide; however, public concern has decreased thanks to successful vaccination efforts and better public health management [1,2]. Since 2020, severe acute respiratory syndrome coronavirus 2 (SARS-Cov-2), the causative agent of COVID-19, has had a high rate of mutation in its spike protein, resulting in more genetic diversity and changes in the prevalence of its variants around the world [3]. The World Health Organization (WHO) and the Centre for Disease Control and Prevention (CDC) have identified several SARS-CoV-2 variants, categorized into variants of concern, variants of interest, and variants of high consequence. These variants include Alpha, Beta, Gamma, and Delta, with the most recent addition being the Omicron variant [4,5]. Most changes have little to no impact on the virus's properties. While some changes may affect the virus's properties as transmissibility and virulence, or the effectiveness of vaccines, therapeutic medicines, diagnostic tools, or other public health and social measures [6].

During this study, the newly emergent strain known as EG.5/Eris, a descendant of Omicron, was the most prevalent SARS-CoV-2 strain of current concern globally [7]. During August 2023, the CDC estimated that EG.5, was responsible for 20.6% of SARS-CoV-2 infections in the United States, which increased to 29.4% at the end of September [8]. The new spike protein mutation of EG.5 may cause immune escape to either previous SARS-CoV-2 infection or the current vaccine. Hence, it may become more infectious with greater severity [9].

Following the emergence of the Omicron variant, new questions arose regarding the duration of vaccine protection and effectiveness against this and other new variants, rather than general concerns about vaccine effectiveness. A study involving 1,285 healthcare workers (HCWs) reported that those who were 55 or older had a strong perception of vaccine ineffectiveness in controlling the Omicron spread [10]. Similarly, another study conducted after the Delta variant outbreak, over 44% of HCWs declined a booster dose, and one-third preferred a new messenger ribonucleic acid (mRNA) vaccine specifically for the new variants [11]. These concerns affected both vaccination rates and the mental health of HCWs. Additionally, it was reported that 66% of HCWs experienced varying degrees of anxiety and depression, with a vaccination rate of only 21.3% attributed to a lack of confidence in the vaccines' ability to prevent infection [12].

Following the initial approval of the COVID-19 vaccine, optimism shifted to concerns over vaccine efficacy against rapidly mutating SARS-CoV-2 strains, raising worries about potential ineffectiveness [13,14]. Concerns have also been raised regarding how vaccine-induced immunity might drive the evolution of the spike protein, potentially impacting the effectiveness of current vaccines [15]. Updated COVID-19 booster shots targeting the XBB.1.5 variant from Moderna, Novavax, and Pfizer aim to enhance protection against new strains. Antivirals like Paxlovid remain effective, and current vaccines and tests still work against emerging variants. Australia has approved the XBB.1.5 vaccine for use in its national program, citing slightly improved protection [16]. However, usual protective precautions may remain crucial, especially with at-risk patients such as the elderly, immunocompromised, and those with chronic disease [17].

Vaccine confidence, as defined by the CDC, denotes the belief in the safety, efficacy, and essential nature of vaccinations within a reliable healthcare system [18]. The level of trust in vaccines varies among individuals and populations, impacting acceptance, utilization, promotion, hesitancy, and rejection. Since the development of the first COVID-19 vaccination, public worries regarding vaccine effectiveness and safety have existed [19]. Vaccine acceptance and confidence are critical determinants of vaccine hesitancy (VH) influencing vaccine uptake, achievement of national immunization targets, and the vulnerability to outbreaks of vaccine-preventable diseases [20,21]. The WHO has clarified VH as one of the ten global health security threats in 2019 that need to be measured and handled by countries [20]. Measuring COVID-19 vaccine confidence would highlight specific concerns affecting an individual's or a community's level of trust toward the received vaccination. Consequently, addressing public concerns through targeted communication and educational campaigns may increase vaccination uptake, reduce transmission, support herd immunity, and ultimately lower SARS-CoV-2 prevalence, easing the burden on overstretched healthcare systems, especially in developing countries [22].

According to the Diagnostic and Statistical Manual of Mental Disorders, Fifth Edition (DSM-5), fear is an emotional response to real or perceived threats, serving a vital role in survival by triggering protective behaviors. However, excessive fear and anxiety can impair decision-making and negatively affect mental health, potentially contributing to conditions such as anxiety disorders, depression, and suicidal ideation [23–25]. Individuals' personal experiences of the pandemic and exposure to misinformation through media may further amplify their feelings of fear and anxiety [26]. Research from various countries indicates that the COVID-19 pandemic has adversely affected mental health, especially for those with pre-existing conditions and limited social support, leading to heightened anxiety, depression, suicidal thoughts, sleep issues, eating disorders, and reduced quality of life. [27,28], and the same for the frontlines HCWs [29]. A study on 2,336 HWCs reported an increasing rate of severe COVID-19 fear from 9% to 15% [30]. Besides impacting their well-being, fear also has a detrimental impact on their occupational outcomes. This finding was reported in other studies in different regions as well [31,32]. Addressing HCWs' fear is crucial for the optimization of their professional performance in preventing outbreaks and as a resource for patient and public health education and awareness [33]. This study aimed to assess

the fear of infection related to new COVID-19 variants and evaluate confidence in the received vaccines among HCWs from different countries, given their crucial role in health education and public awareness.

## Methodology

### Study design and setting

A global outreach cross-sectional study was performed involving healthcare workers from diverse countries and regions around the world, conducted between November 1 and December 5, 2023. These regions included the Eastern Mediterranean Region (EMR) with countries such as Saudi Arabia, Yemen, Libya, Syria, Palestine, the United Arab Emirates, Iraq, and Egypt. The African region (AFR) included Ethiopia, Cameroon, Nigeria, Uganda, and Ghana. The Western region was represented by Germany and Italy, while other parts of the world, including Asia and Latin America, were represented by India, Bangladesh, Mexico, and Mongolia. The data was collected through an anonymous online survey and face-to-face distribution of hard copies of the survey. The online survey was uploaded to Google Forms, distributed through social media platforms (Facebook and Twitter), and sent via email, WhatsApp, and Telegram. The printed-hard copy of the study questionnaire was used to target HCWs who don't have internet access.

### Study population

The current study recruited participants who fulfilled the following inclusion criteria; (1) being HCWs including physicians, dentists, pharmacists, nurses, physical therapists, and administrators in the health services from different countries worldwide, (2) of either sex, (3) being ≥ 18 years old, (4) having a smartphone or computer with access to the internet, and accepted to participate in the study. The co-author (AG) was responsible for recruiting collaborators from the selected countries via the Global Researcher Club (GRC), an international, voluntary, and non-profit scientific research network.

### Sample size and sampling methods

The sample size was calculated using G*power version 3.1, assuming that the estimated proportion of fear toward the new COVID-19 variants was 50%, a power of 95%, and a margin of error of 5%, the size effect of 0.06 (based on the previous study found that 50.6% of Chinese suffered from fear regarding Omicron) [34], the minimum required sample size was 902. By considering the non-response rate of 35%, the sample size increased to 1387. We multiplied by 4 to compensate for stratification (Western region, EMR, African region, and Others). Ultimately, a total of 5,843 HCWs completed the survey, exceeding the minimum requirement and ensuring adequate representation and statistical power across the strata. The participants were recruited using convenience and snowball sampling approaches from HCWs of different countries working at either public or private hospitals and clinics.

### Data collection tools

A self-administered, anonymous questionnaire of five sections was created in English and Arabic to collect the data. *The first section* was about sociodemographic and health-related condition data (i.e., age, gender, marital status, country, place of residence, specialty, any mental health problems). Mental health problems were assessed using self-reported responses to specific questions on the presence of stress, anxiety, sleep disorders, and obsessive-compulsive disorder (OCD). While these were not measured using standardized diagnostic tools, the questionnaire captured participants' acknowledgement of existing conditions, based on their own or a healthcare provider's diagnosis. *The second one* was about the history of previous SARS-CoV-2 infections (i.e., the time since the last infection, self-reported symptoms). *The third section* was about vaccination (i.e., types of vaccine received, number of doses, and post-vaccination side effects). *The fourth section* assessed fear level by the Fear of COVID-19 Scale (FCS), a valid 7-item scale using a 5-point Likert scale ranging from 1 (strongly disagree) to 5 (strongly agree). The total score ranges from 7 to 35, where higher scores

indicate a greater level of fear related to COVID-19 infection [35,36]. The cutoff point for FCS was 17.5 with a sensitivity of 55.1% and specificity of 49.6%, an area under the curve (AUC) of 0.52 (95% confidence interval (CI) = [0.50–0.53], p-value < 0.001). Therefore, those with a total score ≥17.5 were considered to have extreme fear, while those <17.5 were normal [37]. The internal consistency of FCS, as assured by Cronbach's alpha, was (α = 0.912). *The last section* identified confidence level in COVID-19 vaccines using Arabic tool for assessment of post-vaccination confidence in COVID-19 vaccines (ARAB-VAX-CONF) through three domains: assessment of confidence in vaccine effectiveness (8 items), assessment of confidence in vaccine safety (4 items), assessment of confidence in the healthcare system (4 items) [38]. Each item was assessed using a 5-point Likert scale ranging from 1 (strongly disagree) to 5 (strongly agree). Overall confidence was assessed by calculating the composite mean score from the participants' responses to different confidence items, which were rated on a scale from 1 to 5. Participants with a mean composite score below 2.5 were categorized as having poor confidence, those with a score between 2.5 and 3.49 were deemed to have intermediate confidence, and participants with a mean composite score ranging from 3.5 to 5.0 were classified as having good confidence [39]. This scale was developed in the Arabic language. It was translated and adapted cross-culturally, with the forward translation into English by a bilingual professional translator, followed by a backwards translation to check that the meaning of the items was retained. An expert committee composed of two public health professionals and a research methodologist reviewed the clarity of the format and appropriateness of the content, and the necessary adjustments were made based on their recommendations. In addition, the preliminary English version of the questionnaire was pilot tested with a sample of 100 English-speaking individuals to check the clarity of the questions and to estimate the time needed to complete the questionnaire, with no major changes to the instruments. In addition, the reliability of the English version was assessed by Cronbach's alpha (α = 0.802).

## Data collection plan

Before starting data collection, a pilot study was conducted to assess the feasibility, clarity, response rate, and completion time of the questionnaire, besides the accessibility of the online tool by requesting each collaborator to provide at least two responses. There was a cover page to explain the purpose of the study and instructions on how to respond to the questions. Based on the piloted population feedback, we had a response rate of 65%, and some minor edits were made to improve the flow and comprehensibility of the questions. The questionnaire needed 11–15 minutes to be completed. All participants involved in the pilot study were excluded from the final analysis.

## Ethical considerations and approval

The study was approved by the Ethics Committee of the Faculty of Medicine, Alexandria University, Egypt (IRB number: 00012098). The study was conducted following the ethical standards of the 1964 Declaration of Helsinki and its later amendments or comparable ethical guidelines [40]. All participants were informed that their participation was voluntary, and informed written consent was obtained by answering the first question before starting the survey ("to agree" or "not to agree") to participate in the study. Participants did not receive any incentive in return for their participation. Responses were saved in a password-protected computer accessible only to the lead investigator to ensure data confidentiality.

## Data management and analysis

The data were collected, reviewed, and then fed to Statistical Package for Social Sciences (SPSS) version 27 (Armonk, NY: IBM Corp). Numerical variables were described by the mean and standard deviation (SD), whereas categorical variables were described by number (No) and percentage (%). A Chi-square test was used to assess the association between the categorical variables, and the responses were categorized according to receiving the COVID-19 vaccination. An analysis of variance (ANOVA) test was performed to compare the differences between the means of more than two

groups. Tukey's honest significant difference (HSD) post hoc test was used to determine significant differences between multiple groups after the ANOVA test. To estimate associations between dependent and independent variables, univariate logistic regression was used to calculate crude odds ratios (CORs) with 95% CI. Multiple stepwise logistic regression was used to examine the association between dependent and independent variables, quantified using adjusted odds ratios (AORs) with 95% CIs. Two distinct models were developed: the first model identified key determinants influencing fear of COVID-19 emerging variants, while the second model assessed factors shaping confidence in the received vaccine. A p-value < 0.05 was considered statistically significant.

## Results

### Respondents' sociodemographic characteristics

A total of 5843 eligible HCWs completed the survey; their mean age was 32.1 ± 10.8 years, 42.5% were from the EMR, 56.6% were females, 59.1% were single, 76.0% resided in urban regions/cities, 34.6% had a family size of five or more, 48.3% had just enough income, 40.1% were practicing medicine, and 43.6% of participants highest qualification was the bachelor's degree (Supplementary Table 1 in S1 File).

### Medical, mental history, and personal habits of the studied health care workers

The most reported health problems were hypertension (10.4%), respiratory diseases (5.7%), diabetes mellitus (DM) (4.6%), cardiac disease (3.8%), and immunological disease (3.5%), while 45.7% had no chronic health problems. As for mental health problems, stress was reported by 35.3%, followed by anxiety (23.7%), sleep disorders (14.9%), and OCD (4.1%). Regarding smoking habits, 11.2% were current smokers, and most (82.7%) were non-smokers (Supplementary Table 2 in S1 File).

### History of COVID-19 infection among the studied healthcare workers

Among the surveyed HCWs, 30.2% reported a family history of SARS-CoV-2 infection, and 45.7% had been infected themselves. Of those infected (n = 3,291), 81.6% reported that their infection occurred over a year ago. Symptoms were mild in 33.7% of cases and moderate in 46.0%; 4.9% required hospitalization, and 1.5% were admitted to the ICU (Fig 1).

Nearly three-quarters (72.7%) of the surveyed HCWs were vaccinated. Among them, 29.6% received only the primary series, 19.6% received one booster, 16.6% received two boosters, and 6.9% received more than two boosters. Notably, 67.2% reported being obligated to get vaccinated, primarily due to work requirements (68.8%), travel (34.5%), educational demands (31.3%), family pressure (24.8%), and to access government facilities (24.3%). The most commonly received vaccines were Pfizer (40.0%), AstraZeneca (36.8%), and Sinopharm (14.3%). Common side effects included injection site pain (52.9%), fever (38.1%), headache (32.0%), flu-like symptoms (24.6%), and myalgia (23.4%) (Table 1).

### HCWs' fears and perceptions toward the SARS-CoV-2 new emerging variants infection

Exact 30.4% of the participants were most afraid of COVID-19 emerging variants, 28.4% said that it makes them uncomfortable to think about COVID-19, 25.2% were afraid of losing their life because of COVID-19, 22.2% became nervous or anxious when watching news and stories about COVID-19 on social media, and 13.4% reported that their heart rates increases when they thought about getting COVID-19. Details of responses to each item of the FCS (Supplementary File). The overall mean FCS score was 17.1 ± 6.4 out of 35, the lowest mean fear score was among respondents from the EMR (14.6 ± 5.0) followed by the Western region (15.7 ± 4.4) and the highest fear score was among respondents in the other regions (20.4 ± 7.3), (p = 0.002) (Table 2). There was a statistically significant difference in fear score between males and females HCWs across the studied regions, as well (Table 2, Supplementary Figure 1 in S1 File).

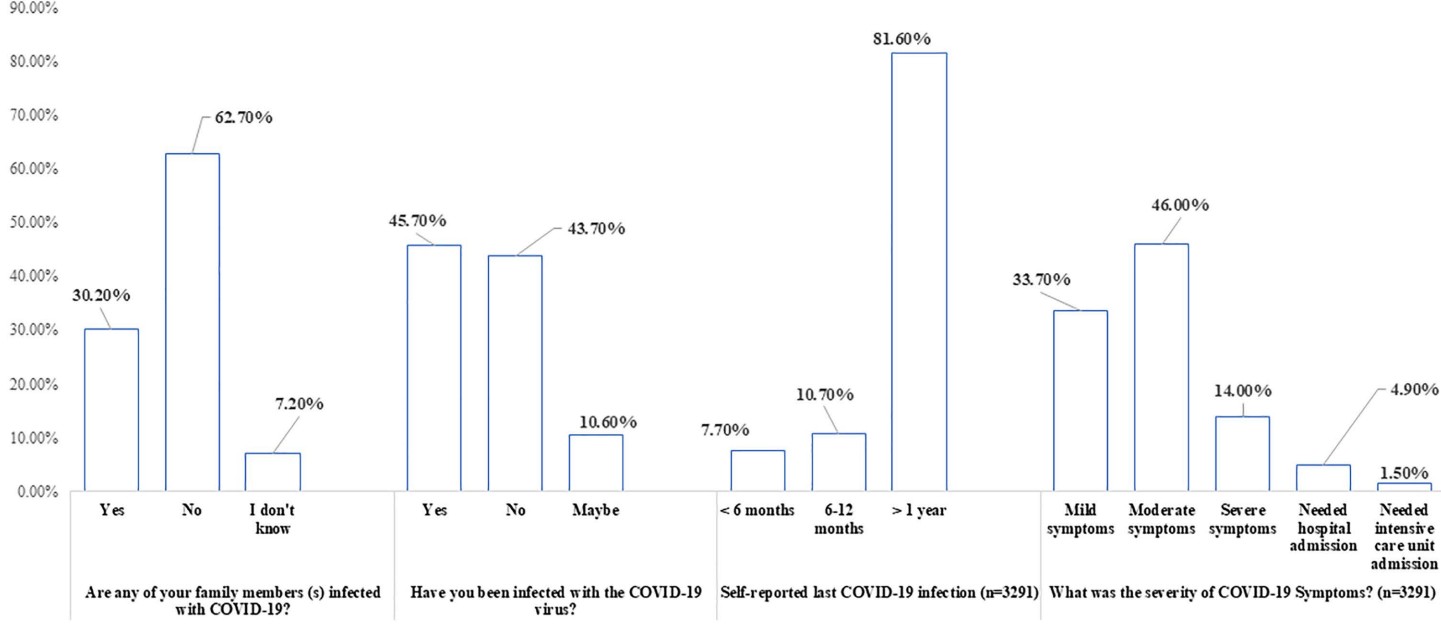

**Fig 1. History of SARS-CoV-2 infection among the studied healthcare workers (n = 5843).**

Fig 2 shows that 59.5% of the respondents had a normal (ordinary) level of fear, but 40.5% were in extreme fear of catching infection with SARS-CoV-2 emerging variants.

Bivariate analysis revealed that extreme fear was reported among African countries (64.6%), divorced/ widow (48.9%), those who had a family member of four (41.6%), those were without enough monthly income and in debit (54.4%), nursing (50.3%), and those who didn't suffer from any chronic disease (48.5%). The place of residence was also significantly associated with fear level ($p < 0.05$) (Table 3).

Multivariable analysis revealed that being married HCWs exhibited lower odds of extreme fear [AOR: 0.8; 95% CI (0.7–0.9)], while divorced or widowed workers showed slightly higher odds [AOR: 1.3; 95% CI (1.0–1.8)]. Place of residence indicated that rural and desert/mountain residents had higher odds of extreme fear [AOR: 1.6; 95% CI (1.4–1.8)] and [AOR: 2.5; 95% CI (1.6–4.0)], respectively. Family size also affected fear levels, with smaller families (two and three) showing lower odds [AOR: 0.63; 95% CI (0.5–0.78)] and [AOR: 0.62; 95% CI (0.51–0.72)] respectively. Those without enough and in debt and those with not enough income level had more than two times higher odds of extreme fear [AOR: 2.5; 95% CI (2.2–3.1)] and [AOR: 2.4; 95% CI (2.0–2.8)], respectively. Having chronic diseases increased the odds of extreme fear [AOR: 1.2; 95% CI (1.1–1.4)]. Being dentist [AOR: 0.51; 95% CI (0.33–0.75)], physiotherapist [AOR: 0.80; 95% CI (0.60–0.97)], physician [AOR: 0.77; 95% CI (0.61–0.92)], pharmacist [AOR: 0.75; 95% CI (0.55–0.92)] were associated with lower fear (Table 3).

## Confidence in COVID-19 vaccines among HCWs who received the vaccines (n=4251)

Details on confidence regarding vaccine effectiveness, vaccine safety, and confidence in health care. Overall, among those who received the COVID-19 vaccination, 41.0% showed good confidence about the vaccine, while 4.7% showed poor confidence. The highest confidence was reported among AFR region HCWs (65.7% rated as good), followed by Western region's HCWs (49.9%), while nearly two-thirds (68.4%) of EMR's HCWs had intermediate confidence towards the received COVID-19 vaccine (Fig 3).

**Table 1. COVID-19 vaccination status and associated side effects among the studied healthcare workers (n = 5,843).**

| Vaccination | | No | % |
|---|---|---|---|
| **How many COVID-19 vaccination doses did you receive?** | I didn't receive the vaccine | 1592 | 27.3 |
| | The primary series | 1727 | 29.6 |
| | One booster | 1148 | 19.6 |
| | Two boosters | 970 | 16.6 |
| | More than two boosters | 406 | 6.9 |
| **Were you obliged to take the vaccination? (n = 4251)** | Yes | 2857 | 67.2 |
| | No | 1394 | 32.8 |
| **If obligated, mention the cause# (n = 2857)** | Work requirement | 1966 | 68.8 |
| | Travel prerequisite | 986 | 34.5 |
| | Educational requirements | 893 | 31.3 |
| | Family pressure | 709 | 24.8 |
| | Entry to governmental facilities | 695 | 24.3 |
| | Peer pressure | 311 | 10.9 |
| | Other | 401 | 14.0 |
| **What was the type of the received vaccine?# (n = 4251)** | Pfizer | 1702 | 40.0 |
| | AstraZeneca | 1565 | 36.8 |
| | Sinopharm | 610 | 14.3 |
| | Moderna | 411 | 9.7 |
| | Johnson & Johnson | 364 | 8.6 |
| | Sinovac | 261 | 6.1 |
| | Sputnik | 256 | 6.0 |
| | I don't know/remember | 215 | 5.1 |
| **What were the side effects of the received vaccine? (n = 4251) #** | No side effects | 1579 | 37.1 |
| | Pain at the site of injection | 2248 | 52.9 |
| | Fever | 1619 | 38.1 |
| | Headache | 1362 | 32.0 |
| | Flu-like symptoms | 1045 | 24.6 |
| | Myalgia | 994 | 23.4 |
| | Bone pain | 730 | 17.2 |
| | Allergy | 253 | 6.0 |
| | Others | 266 | 6.3 |

#This is a multiple-response question.

**Table 2. The overall fear score of emerging COVID-19 variants among respondents from different regions.**

| | | COVID-19 fear score | | p-value# |
|---|---|---|---|---|
| | | **Mean** | **SD** | |
| **World Regions** | Western Region [a] | 15.7 | 4.4 | 0.002* |
| | Eastern Mediterranean Region [a] | 14.6 | 5.0 | |
| | African countries [b] | 19.7 | 6.8 | |
| | Others [c] | 20.4 | 7.3 | |
| | Total | 17.1 | 6.4 | |

#ANONA test *Statistically significant (p < 0.05). a: statistically significant differences compared with all regions. b: statistically significant differences compared with all regions except others. c: statistically significant differences compared with all regions except African countries.

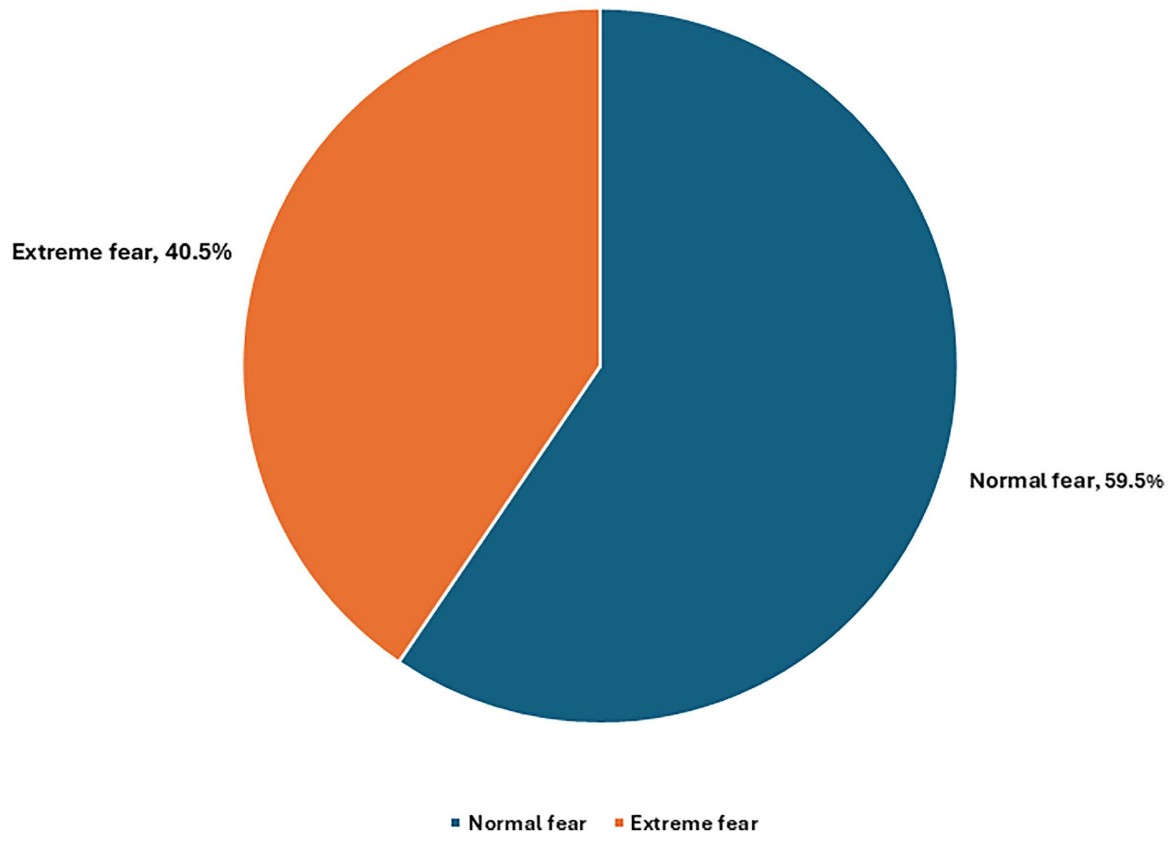

**Fig 2. Healthcare workers' fear of infection with emerging variants of SARS-Cov-2.**

Bivariate analysis revealed that, except for place of residence and having chronic diseases, there were significant associations between all the studied variables and confidence level in the received vaccine. Multivariable analysis revealed that higher confidence was reported among 30–39 age group [AOR: 1.4; 95% CI (1.1–1.8)], ≥40 age group [AOR: 1.8; 95% CI (1.4–2.3)], rural and other residence [AOR: 1.3; 95% CI (1.1–1.5)], males [AOR: 1.3; 95% CI (1.1–1.4)], and small family member of one [AOR: 5.5; 95% CI (4.2–7.2)], two [AOR: 1.5; 95% CI (1.2–1.9)], and three [AOR: 1.3; 95% CI (1.1–1.6)]. On the other hand, having chronic diseases [AOR: 0.82; 95% CI (0.71–0.95)], having mental disorders [AOR: 0.59; 95% CI (0.51–0.69); p < 0.01], high family number (4 number) [AOR: 0.78; 95% CI (0.69–0.89)], personal history of COVID-19 infection [AOR: 0.61; 95% CI (0.53–0.71)], and experiencing side effects of COVID-19 vaccination [AOR: 0.63; 95% CI (0.55–0.72)] were associated with low confidence in the received vaccine (Table 4).

## Discussion

This study aimed to examine the fear of infection from emerging SARS-CoV-2 variants and the level of confidence in received COVID-19 vaccines among HCWs across multiple regions. A substantial proportion of HCWs (40.5%) reported extreme fear of infection with new variants, while only 41.0% expressed good confidence in the vaccines they received. Several demographics, professional, and health-related factors were significantly associated with both fear and vaccine confidence levels.

**Table 3. Determinants of extreme fear of infection with emerging COVID-19 Variants among healthcare workers (n = 5843).**

| Studied variables | | Fear of COVID–19 | | | | p–value# | COR (95% CI) | AOR (95% CI) |
|---|---|---|---|---|---|---|---|---|
| | | Normal | | Extreme fear | | | | |
| | | No | % | No | % | | | |
| World Regions | Western region | 598 | 70.9% | 245 | 29.1% | .001* | 1 | 1 |
| | Eastern Mediterranean region | 1969 | 79.4% | 512 | 20.6% | | 0.63 (0.53-1.15) | 0.61 (0.55-1.13) |
| | African region | 500 | 35.4% | 914 | 64.6% | | 4.4(3.7–5.3) | **3.7 (3.3–4.5)** |
| | Others | 410 | 37.1% | 695 | 62.9% | | 4.1(3.4–5.0) | **3.2(2.6–4.0)** |
| Marital status | Single | 1969 | 57.1% | 1480 | 42.9% | .001* | 1 | 1 |
| | Married | 1393 | 64.4% | 770 | 35.6% | | 0.7(0.6–0.8) | **0.8(0.7–0.9)** |
| | Divorced/ widow | 114 | 51.1% | 109 | 48.9% | | 1.2(0.9–1.6) | **1.3(1.0–1.8)** |
| Place of residence | Urban/ city | 2834 | 63.9% | 1604 | 36.1% | .001* | 1 | 1 |
| | Rural/ village | 589 | 46.5% | 679 | 53.5% | | 2.0(1.7–2.3) | **1.6(1.4–1.8)** |
| | Desert/ mountain | 40 | 42.6% | 54 | 57.4% | | 2.3(1.5–3.6) | **2.5(1.6–4.0)** |
| | Others£ | 14 | 32.6% | 29 | 67.4% | | 3.6(1.9–6.9) | **2.2(1.1–4.5)** |
| Number of family members | 1 | 340 | 48.0% | 368 | 52.0% | .001* | 1 | 1 |
| | 2 | 355 | 59.1% | 246 | 40.9% | | 0.64 (0.51-0.79) | **0.63 (0.50-0.78)** |
| | 3 | 558 | 60.4% | 366 | 39.6% | | 0.61 (0.49-0.73) | **0.62 (0.51-0.72)** |
| | 4 | 927 | 58.4% | 660 | 41.6% | | 0.65 (0.55–0.78) | 0.76 (0.63-1.0) |
| | ≥5 | 1297 | 64.1% | 726 | 35.9% | | 0.52 (0.44–0.62) | 0.83 (0.67–1.0) |
| Level of monthly income | Not enough and in debt | 325 | 45.6% | 388 | 54.4% | .001* | 2.6 (2.2-3.2) | **2.5 (2.2-3.1)** |
| | Not enough | 624 | 46.9% | 707 | 53.1% | | 2.5 (2.1-2.9) | **2.4 (2.0-2.8)** |
| | Just enough | 1856 | 65.8% | 966 | 34.2% | | 1.2 (1.0-1.4) | 1.2 (0.9-1.14) |
| | Enough and saving | 672 | 68.8% | 305 | 31.2% | | 1 | 1 |
| Field of practice in healthcare | Administrative | 177 | 56.0% | 139 | 44.0% | .001* | 1 | 1 |
| | Dentistry | 331 | 72.4% | 126 | 27.6% | | 0.48 (0.31-0.74) | **0.51 (0.33-0.75)** |
| | Health and rehabilitation sciences "Physiotherapy" | 147 | 60.7% | 95 | 39.3% | | 0.82 (0.62-1.0) | **0.80 (0.60-0.97)** |
| | Medicine | 1441 | 61.6% | 899 | 38.4% | | 0.79 (0.60-0.92) | **0.77 (0.61-0.92)** |
| | Nursing | 483 | 49.7% | 488 | 50.3% | | 1.29 (0.86-12.32) | 1.17 (0.83-2.21) |
| | Pharmacy | 537 | 63.3% | 311 | 36.7% | | 0.74 (0.59-0.96) | **0.75 (0.55-0.92)** |
| | Others | 361 | 54.0% | 308 | 46.0% | | 1.09 (0.86-2.25) | **1.04 (0.88-2.23)** |
| Chronic diseases | Yes | 1843 | 69.1% | 825 | 30.9% | .001* | 2.1 (1.8–2.3) | **1.2(1.1–1.4)** |
| | No | 1634 | 51.5% | 1541 | 48.5% | | 1 | 1 |

#Chi square test *Statistically significant (p < 0.05). £ others refers non-standard or unlisted living areas such as refugee camps, temporary housing, or remote settlements. COR: Crude odds ratio; AOR: Adjusted odds ratio; CI (Confidence interval)

### Fear of emerging COVID-19 variants among HCWs

SARS-CoV-2 continues to mutate, with the EG.5 Omicron lineage being the latest variant of interest at the time of this study [41]. As frontline responders, HCWs are directly exposed to evolving risks, which may exacerbate psychological stress. Our findings revealed that the mean score on the FCS was 17.1 ± 6.4, comparable to previous research indicating heightened anxiety and fear among HCWs during the pandemic [42–44]. The neural substrates underlying fear responses and vaccine confidence among HCWs may help understand the psychological and neurobiological factors influencing vaccine acceptance. Fear and risk perception are rooted in complex neurocognitive processes involving key brain regions such as the amygdala, insula, and prefrontal cortex. These areas govern emotional responses and decision-making under

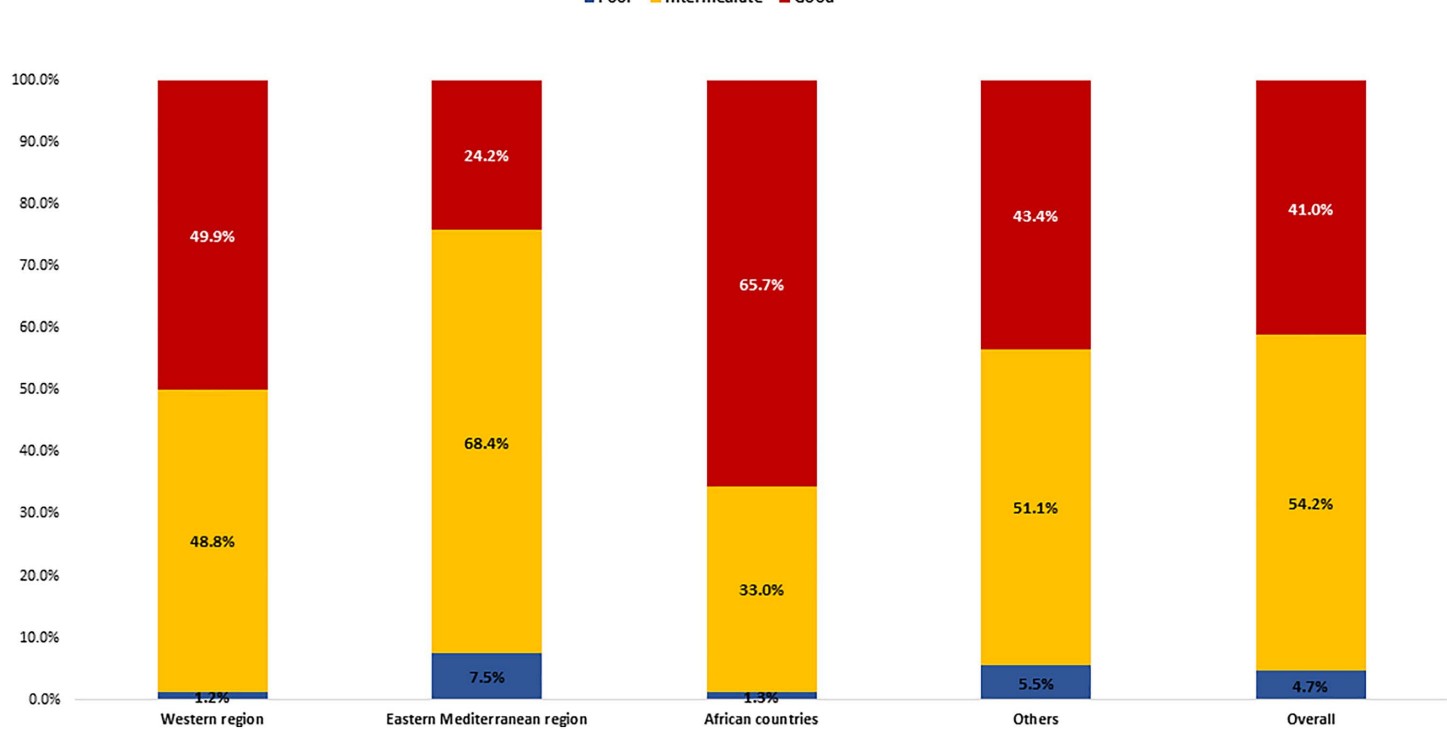

**Fig 3. Healthcare workers' confidence in the received COVID-19 vaccine by region.**

uncertainty. The extreme fear of infection with SARS-CoV-2 emerging variants observed among many HCWs in our study could be partially explained by the activation of these regions in response to continuous exposure to pandemic-related stressors, uncertainty about variant severity, and personal vulnerability. This heightened fear may, in turn, affect HCWs' confidence in vaccines, compliance with preventive measures, and even their willingness to remain in the workforce during pandemics.

Gender-based differences in fear were observed across the studied regions; however, gender was not a significant predictor of fear in the multivariable analysis. Fear scores were higher among females in the EMR and Western regions, while males exhibited more fear in Africa and other global regions. These findings are consistent with the literature suggesting that women are generally more susceptible to anxiety-related disorders [45–48], although some studies reported no gender difference [49–51]. Sociocultural factors and differing resilience patterns may partially explain these variations [51].

Marital status was a significant determinant of fear score. Higher fear was reported among divorced or widowed individuals. In the same vein, many studies found higher prevalence of happiness and less fear and worry among married individuals during COVID-19 [52–54]. Being married may provide emotional and instrumental support during illness, which could help reduce fear and anxiety related to COVID-19 infection and its emerging variants [53]. In the current study, having a small family was significantly associated with lower levels of fear. This may reflect stronger social support systems and perceived stability among married individuals, which can serve as a psychological buffer against fear and anxiety.

Insufficient income was significantly associated with the fear level. Beside VH [55], insufficient income was significantly associated with many mental health problems during COVID-19, like insomnia [56], quality of life [29], mental health, and family relationships [57]. On the other hand, a study in Iran did not find a significant association between fear and income

**Table 4. Multiple stepwise logistic regression model for determinants of healthcare workers' confidence in the received COVID-19 vaccine.**

| Factors | COVID–19 vaccine confidence level | | | | | | p–value# | COR (95% CI) | AOR (95% CI) |
|---|---|---|---|---|---|---|---|---|---|
| | Poor | | Intermediate | | Good | | | | |
| | No | % | No | % | No | % | | | |
| **Age in years** | | | | | | | <.001* | | |
| 18–24 | 43 | 4.0% | 656 | 60.7% | 381 | 35.3% | | 1 | 1 |
| 25–29 | 62 | 5.4% | 657 | 57.3% | 428 | 37.3% | | 1.0 (0.9-1.3) | 1.2 (0.89-1.5) |
| 30–39 | 46 | 4.5% | 512 | 50.3% | 460 | 45.2% | | 1.5 (1.3-1.8) | **1.4 (1.1-1.8)** |
| 40+ | 49 | 4.9% | 481 | 47.8% | 476 | 47.3% | | 1.6 (1.3-2.0) | **1.8 (1.4-2.3)** |
| **Gender** | | | | | | | <.001* | | |
| Male | 89 | 5.0% | 887 | 49.6% | 812 | 45.4% | | 1.4 (1.2–1.5) | **1.3 (1.1–1.4)** |
| Female | 111 | 4.5% | 1419 | 57.6% | 933 | 37.9% | | 1 | 1 |
| **Place of residence** | | | | | | | .077 | | |
| Urban/ city | 157 | 4.8% | 1814 | 55.1% | 1321 | 40.1% | | 1 | 1 |
| Rural/ Others | 43 | 4.5% | 492 | 51.3% | 424 | 44.2% | | 1.2 (1.0–1.4) | **1.3 (1.1–1.5)** |
| **Number of family members** | | | | | | | <.001* | | |
| 1 | 7 | 2.1% | 84 | 25.2% | 242 | 72.7% | | 5.6 (4.3-7.4) | **5.5 (4.2-7.2)** |
| 2 | 17 | 4.6% | 197 | 52.8% | 159 | 42.6% | | 1.6 (1.3-2.0) | **1.5 (1.2-1.9)** |
| 3 | 38 | 4.9% | 430 | 55.8% | 303 | 39.3% | | 1.4 (1.1-1.7) | **1.3 (1.1-1.6)** |
| 4 | 60 | 4.5% | 696 | 51.9% | 584 | 43.6% | | 1.02 (0.65-1.9) | **0.78(0.69-0.89)** |
| ≥5 | 78 | 5.4% | 899 | 62.7% | 457 | 31.9% | | | 1 |
| **Having chronic diseases** | | | | | | | .384 | | |
| No | 93 | 4.6% | 1071 | 53.2% | 850 | 42.2% | | 1 | 1 |
| Yes | 107 | 4.8% | 1235 | 55.2% | 895 | 40.0% | | 0.91 (0.81–1.0) | **0.82 (0.71–0.95)** |
| **Having a mental health problem** | | | | | | | <.001* | | |
| No | 108 | 4.8% | 1086 | 47.9% | 1074 | 47.4% | | 1 | 1 |
| Yes | 92 | 4.6% | 1220 | 61.5% | 671 | 33.8% | | 0.57 (0.50–0.64) | **0.59 (0.51–0.69)** |
| **Having family member (s) infected with COVID–19?** | | | | | | | <.001* | | |
| Yes | 71 | 4.5% | 904 | 57.4% | 599 | 38.1% | | 0.82 (0.72–0.93) | **0.96 (0.83–1.1)** |
| No | 119 | 5.0% | 1303 | 54.6% | 964 | 40.4% | | 1 | 1 |
| I don't know† | 10 | 3.4% | 99 | 34.0% | 182 | 62.5% | | | |
| **History of COVID–19 virus infection** | | | | | | | <.001* | | |
| Yes | 127 | 5.6% | 1316 | 57.6% | 841 | 36.8% | | 0.69 (0.61–0.78) | **0.61 (0.53–0.71)** |
| No | 45 | 2.9% | 759 | 48.3% | 768 | 48.9% | | 1 | 1 |
| Maybe† | 28 | 7.1% | 231 | 58.5% | 136 | 34.4% | | | |
| **Had side effects of the COVID-19 vaccination** | | | | | | | <.001* | | |
| No | 64 | 4.1% | 710 | 45.0% | 805 | 51.0% | | 1 | 1 |
| Yes | 136 | 5.1% | 1596 | 59.7% | 940 | 35.2% | | 0.52 (0.46–0.59) | **0.63 (0.55–0.72)** |

#Chi square test. COR: Crude odds ratio; AOR: Adjusted odds ratio; CI (Confidence interval); *P<0.05 (significant). † merged with no.

level [58]. This can be attributed to the fact that individuals with lower income levels are more likely to be employed in sectors that were heavily affected by COVID-19. Furthermore, the pandemic has had a disproportionately negative impact on underprivileged families. Financial insecurity, a key stressor, can create a tense family environment and heighten the risk of conflicts [59].

Residing in a non-urban area significantly increases the fear among the studied HCWs. A similar finding was reported in several studies [60,61]. This may be due to the disparity in the spread of COVID-19 between urban and rural areas.

This gap continues to influence healthcare access, risk perception, and preventive behaviors across different populations [62]. Regionally, HCWs from the African Region reported the highest fear scores. These patterns may reflect regional differences in pandemic experiences, healthcare infrastructure, or cultural perceptions of health and risk, as previously reported in EMR-focused studies [63].

Working as a physician, dentist, pharmacist, or physiotherapist was associated with lower levels of fear. HCWs in clinical roles may have greater access to accurate medical information and a better understanding of transmission dynamics, reducing uncertainty and fear. Their routine exposure to infectious diseases might also contribute to a higher baseline of preparedness and resilience, further lowering perceived threat from emerging variants.

## Confidence in the received COVID-19 vaccines

In our study, only 41% of HCWs reported good confidence in the vaccines they received, with significant variation across regions. Confidence was evaluated across three domains: vaccine effectiveness, healthcare system trust, and vaccine safety. HCWs generally expressed strong beliefs in the vaccine's effectiveness, agreeing with statements like "Vaccination can save many lives" and "Vaccination improves immune protection." However, some expressed a belief that infection might be preferable to vaccination, reflecting lingering concerns about side effects or efficacy. This mirrors global patterns of VH, with studies in the United Kingdom and the United States indicating that a notable proportion of HCWs expressed uncertainty or skepticism about vaccine safety and effectiveness [64–67]. Trust in the healthcare system also influences confidence. HCWs who trusted government-endorsed vaccination programs and regulatory bodies were more likely to exhibit vaccine confidence, consistent with findings from Mohammed et al. and Kreps & Kriner [68,69]. Vaccine safety was another concern, over a quarter of HCWs were worried about side effects, and some felt like "guinea pigs" due to the speed of vaccine development. Similar sentiments have been reported among nursing students and the general population [48,70].

Confidence levels varied substantially by region in bivariate analysis but not in multivariable analysis. HCWs from African and Western regions demonstrated the highest levels of confidence, while those from the EMR mostly reported intermediate confidence. Interestingly, the EMR also had the lowest fear scores, suggesting a possible inverse relationship between fear and confidence.

Demographically, greater vaccine confidence was linked to older age groups, particularly those aged 30–39 and 40 and above, as well as male gender, rural residence, and smaller family sizes. These patterns align with findings from other studies that indicate higher confidence among older healthcare workers and males. This may be attributed to a greater perceived vulnerability to severe outcomes from COVID-19 or variations in media consumption habits [71–74]. Moreover, older HCWs may perceive themselves as at greater risk of severe illness, which can increase motivation to accept vaccination. Rural residents may have different health beliefs or greater trust in public health programs, especially when access to healthcare is limited. Meanwhile, HCWs with smaller families may experience fewer competing priorities or caregiving burdens, allowing them to engage more confidently with vaccination campaigns.

## Interplay between fear of new SARS-CoV-2 variant infection and confidence in the received vaccine

While fear of infection might be expected to motivate vaccine acceptance, our findings suggest a more complex dynamic. HCWs with chronic diseases reported greater fear and low confidence, while those who had mental health conditions exhibited less vaccine confidence. In fact, prior COVID-19 infection and post-vaccination side effects were associated with lower vaccine confidence, contrary to what might be assumed based on personal experience with the disease. Interestingly, this differs from findings in the general population. Dumitra et al. [75] reported that vaccine refusal among the public was often driven by a belief in natural immunity or a perceived lack of necessity. In contrast, our study indicates that HCWs, despite their firsthand exposure to the disease, may lose confidence after observing or experiencing breakthrough infections or adverse effects. This suggests that their clinical knowledge and experience may make them more critical of vaccine performance.

It's important to note that since the global rollout of COVID-19 vaccines, public and professional confidence has evolved. This confidence has been influenced not only by increasing real-world data on vaccine safety and effectiveness but also by changing narratives regarding waning immunity, the duration of protection, and the emergence of immune-evasive variants. These factors could partially explain the observed moderate-to-low confidence levels in some groups. Furthermore, the availability of more effective treatments and clinical protocols for managing COVID-19, including antivirals, monoclonal antibodies, and improved ICU care, may have contributed to mitigating fear of severe illness or death, especially among healthcare workers who have observed these improvements firsthand.

Moreover, another important factor shaping HCWs' vaccine perceptions is the evolving nature of the pandemic itself. Following the peak of the Delta wave, subsequent Omicron lineages (e.g., BA.5, XBB, EG.5) were associated with progressively milder clinical presentations and reduced hospitalization rates. This observed decrease in severity may have influenced a shift in HCWs' behavior—lowering fear levels and contributing to reduced uptake or enthusiasm for booster doses. A sense of futility or "booster fatigue" may have emerged, particularly as new evidence revealed the relatively short duration of protection conferred by additional booster doses. This phenomenon is supported by Savulescu et al. [76], who argue that waning immunity and the relatively short-lived protection of boosters may contribute to vaccine fatigue and declining confidence among HCWs, especially in the context of decreasing disease severity. These evolving attitudes emphasize the importance of transparent, science-based communication tailored to healthcare professionals, who are both recipients and promoters of vaccination. These insights underscore the complexity of vaccine decision-making. While laypeople may decline vaccines due to skepticism or philosophical beliefs, HCWs may become disillusioned due to empirical, lived experience. Addressing these concerns requires tailored messaging that goes beyond safety reassurance, offering transparent explanations of evolving evidence and the rationale for updated recommendations.

## Study limitations and future directions

The current study has some limitations: *first*, mental health problems were self-reported, and we did not use validated screening tools. Future studies employing validated instruments to measure mental health could yield more robust insights. *Second*, we included a small sample size from specific regions (e.g., the Western region and parts of Eastern Asia and Latin America). This may be due to differences in collaborator networks, response rates, and ease of access to HCWs. However, this disparity introduces potential sampling selection biases and limits the generalizability of our findings, so future studies should consider employing proportionate or quota sampling strategies based on global health workforce distribution to improve representativeness. *Third,* the study design, a cross-sectional design, cannot assess the causality between the studied factors and fear of infection with emerging variants of COVID-19 or confidence in the received vaccine. It can only formulate a hypothesis for future research. *Fourth,* the survey did not capture information on whether the received COVID-19 vaccines were monovalent or bivalent, which may limit the interpretation of findings in relation to variant-specific immunity and vaccine confidence. *Finally,* although our study provides insight into how personal COVID-19 infection and vaccine side effects influence HCWs' confidence, we did not assess whether HCWs were directly involved in the care of COVID-19 patients or assigned to isolation/treatment units. This is an important distinction, as direct clinical exposure may intensify fear, shape perceptions of vaccine effectiveness, or reinforce skepticism due to observed breakthrough cases. Prior research has shown that frontline HCWs working in COVID-19 units may experience higher psychological distress and may interpret their experiences through different cognitive frames compared to those in non-COVID units.

## Conclusions and recommendations

Extreme fear of infection with emerging COVID-19 variants was reported by two-fifths of HCWs and showed significant variation across regions and genders. Notably, higher fear levels were observed among HCWs in African and other non-Western regions. Predictors of increased fear included being divorced or widowed, living in rural and desert/mountain

areas, having insufficient income, and having chronic diseases. On the contrary, being married and having an occupation as a dentist, physician, physiotherapist, or pharmacist) were associated with lower odds of extreme fear. Predictors of good vaccine confidence were older age, male sex, having small family members (1–3), and residing in rural areas. Conversely, having a chronic disease, or mental disorders, a personal history of COVID-19 infection, and experiencing side effects of COVID-19 vaccination were associated with lower vaccine confidence. These findings underscore the need for providing targeted educational programs and personalized psychological support and encourage open communication to address these concerns effectively and to dispel uncertainties, particularly regarding efficacy and side effects. Future interventions and policies should be tailored to address regional and demographic variations in COVID-19 emerging variants-related fear and vaccine confidence in the received vaccine, promoting widespread vaccine acceptance among HCWs.

## Supporting information

**S1 Data. Data file COVID-19 variants related Fear.**
(XLSX)

**S1 File. Supplementary file.**
(DOCX)

**S1 Questionnaire. Eg.5 Questionnaire English.**
(DOCX)

**S2 Questionnaire. Inclusivity in global research questionnaire.**
(DOCX)

**S1 Checklist. PLOSOne human subjects research checklist.**
(DOCX)

## Acknowledgments

We would like to thank the study participants for their voluntary participation in this research. The authors extend their appreciation to the Deanship of Research and Graduate Studies at King Khalid University, KSA, for funding this work. (Through Small Research Group under grant number (RGP.1/272/45/ 1445.).

## Author contributions

**Conceptualization:** Mai Hussein.

**Data curation:** Mai Hussein, Assem Gebreal, Ahmed Naeem, Asmaa Mohammed AboElela, Hoda Ali Ahmed Shiba, Vanessa Pamela Salolin Vargas.

**Formal analysis:** Ahmed Naeem, Asmaa Mohammed AboElela, Hoda Ali Ahmed Shiba, Vanessa Pamela Salolin Vargas.

**Investigation:** Mai Hussein, Shehata Farag Shehata, Ahmed A. Mahfouz.

**Methodology:** Mai Hussein, Jargaltulga Ulziijargal, Ibrahim Adel, Shehata Farag Shehata.

**Project administration:** Assem Gebreal, Ibrahim Adel.

**Resources:** Assem Gebreal, Jargaltulga Ulziijargal, Aesha L. E. Enairat, Ibrahim Adel, Bayan Ayash, Omar Alwakaa, Logina Ezz Elarab, Fabio Massimo Oddi, Dennis Brempong.

**Software:** Aesha L. E. Enairat, Bayan Ayash, Omar Alwakaa, Logina Ezz Elarab, Fabio Massimo Oddi, Hala Bakro, Dennis Brempong, Muhereza Morgan Meike.

**Supervision:** Safar Abadi Alsaleem, Ahmed A. Mahfouz, Hala Bakro, Muhereza Morgan Meike, Ramy Mohamed Ghazy.

**Visualization:** Ramy Mohamed Ghazy.

**Writing – original draft:** Mai Hussein, Hoda Ali Ahmed Shiba.

**Writing – review & editing:** Safar Abadi Alsaleem, Ahmed A. Mahfouz, Ramy Mohamed Ghazy.

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
