## [Decision Letter · Decision Letter 0]

11 Sep 2024

Dear Dr. Ghazy,

Thank you for submitting your manuscript to PLOS ONE. After careful consideration, we feel that it has merit but does not fully meet PLOS ONE’s publication criteria as it currently stands. Therefore, we invite you to submit a revised version of the manuscript that addresses the points raised during the review process.

**ACADEMIC EDITOR: No comment**

We look forward to receiving your revised manuscript.

Kind regards,

Stephen Dajaan Dubik, BSc, MPH, MPhil

Academic Editor

PLOS ONE

Journal Requirements:

For additional information about PLOS ONE ethical requirements for human subjects research, please refer to http://journals.plos.org/plosone/s/submission-guidelines#loc-human-subjects-research .

3. Please include a complete copy of PLOS’ questionnaire on inclusivity in global research in your revised manuscript. Our policy for research in this area aims to improve transparency in the reporting of research performed outside of researchers’ own country or community. The policy applies to researchers who have travelled to a different country to conduct research, research with Indigenous populations or their lands, and research on cultural artefacts. The questionnaire can also be requested at the journal’s discretion for any other submissions, even if these conditions are not met.  Please find more information on the policy and a link to download a blank copy of the questionnaire here: https://journals.plos.org/plosone/s/best-practices-in-research-reporting. Please upload a completed version of your questionnaire as Supporting Information when you resubmit your manuscript.

4. In this instance it seems there may be acceptable restrictions in place that prevent the public sharing of your minimal data. However, in line with our goal of ensuring long-term data availability to all interested researchers, PLOS’ Data Policy states that authors cannot be the sole named individuals responsible for ensuring data access (http://journals.plos.org/plosone/s/data-availability#loc-acceptable-data-sharing-methods).

6. Please amend your list of authors on the manuscript to ensure that each author is linked to an affiliation. Authors’ affiliations should reflect the institution where the work was done (if authors moved subsequently, you can also list the new affiliation stating “current affiliation:….” as necessary).

7. Your ethics statement should only appear in the Methods section of your manuscript. If your ethics statement is written in any section besides the Methods, please delete it from any other section.

Reviewers' comments:

Reviewer's Responses to Questions

**Comments to the Author**

1. Is the manuscript technically sound, and do the data support the conclusions?

Reviewer #1: Yes

Reviewer #2: Partly

2. Has the statistical analysis been performed appropriately and rigorously?

Reviewer #1: Yes

Reviewer #2: Yes

3. Have the authors made all data underlying the findings in their manuscript fully available?

Reviewer #1: No

Reviewer #2: Yes

4. Is the manuscript presented in an intelligible fashion and written in standard English?

Reviewer #1: Yes

Reviewer #2: Yes

Reviewer #1: Dear Authors,

This is an interesting paper. I have the following comments

1) Kindly define the outcome terms in methods section - example normal fear/extreme fear. What do they mean? What were their cut-offs? How did you come to that

2) What is good confidence/intermediate/poor confidence

Most of your results and discussion are from these outcomes. It will be useful for the reader to understand it.

Hope these comments are useful

Reviewer #2: This cross-national study titled ‘Cross-National Disparities in Healthcare Workers' Perceptions: Examining Fear of Infection and Confidence in COVID-19 Vaccines in the Context of Emerging Variants’ examined healthcare workers' (HCWs) perceptions of COVID-19 infection fear and vaccine confidence in the context of emerging variants. The research involved 5,843 HCWs from various global regions, with a mean age of 32.1 years. Data was collected through online surveys and face-to-face interviews between November 1 and December 5, 2023, using validated scales to assess fear levels and vaccine confidence. The study sample was diverse, with 42.5% of participants from the Eastern Mediterranean Region, 24.2% from the African Region, 14.4% from Western countries, and 18.9% from other world regions. A majority (72.8%) of participants were vaccinated, with Pfizer, AstraZeneca, and Sinopharm being the most common vaccines received.

Results indicated that a significant portion of HCWs experienced fear related to COVID-19, with 30.4% feeling uncomfortable and 28.4% fearing for their lives. However, 35.3% of participants demonstrated good attitudes and confidence towards the vaccine. Factors associated with higher vaccine confidence included older age, male gender, presence of chronic and mental disorders, larger family size, and previous COVID-19 infection. Conversely, concerns about side effects were linked to lower vaccine confidence. The study revealed regional differences in attitudes towards the COVID-19 vaccine among HCWs, so the authors concluded that targeted educational programs should be implemented to address uncertainties and promote widespread vaccine acceptance among healthcare workers. These findings provide valuable insights for developing strategies to enhance vaccine confidence and address fears among HCWs in different global regions.

I think the idea of this article may be of interest to the readers of Plos One. However, some comments, as well as some crucial evidence that should be included to support the authors’ argumentation, needed to be addressed to improve the quality of the manuscript, its adequacy, and its readability prior to the publication in the present form. My overall judgment is to publish this research article after the authors have carefully considered my suggestions below, in particular reshaping parts of the Introduction and Methods sections by adding more evidence.

Strengths:

• Large, diverse sample size (5,843 healthcare workers) across multiple regions provides good statistical power and cross-national comparisons.

• Use of validated scales to measure fear (FCS) and vaccine confidence (ARAB-VAX-CONF) strengthens the reliability of the findings.

Please consider the following comments:

• The introduction should provide more background on the emergence of new COVID-19 variants and their potential impact on healthcare workers' perceptions and behaviors. In my opinion, this is crucial for understanding the study's relevance.

• A brief review of previous studies on healthcare workers' attitudes towards COVID-19 vaccines and their fears related to infection would be beneficial, as this would help establish the current state of knowledge and identify gaps this study aims to fill. Additionally, the authors could strengthen the rationale for their study by explaining why understanding healthcare workers' perceptions is particularly important in the context of emerging variants.

• The authors may consider incorporating a brief discussion on the potential neural substrates underlying fear responses and vaccine confidence among healthcare workers. Recent neuroimaging studies have identified brain regions involved in processing fear and anxiety, such as the amygdala, insula, and prefrontal cortex, which could be relevant to understanding the neurobiological basis of COVID-19 related fears. Additionally, exploring the neural correlates of decision-making processes related to vaccine acceptance could provide valuable insights. Including this perspective could strengthen the paper's theoretical framework and suggest directions for future research combining psychological and neuroscientific approaches to vaccine hesitancy among healthcare professionals [1-3].

• The manuscript lacks details on how participants were recruited across different regions. More information is needed on the sampling method and how representativeness was ensured.

• While they mention both online surveys and face-to-face interviews were used, authors have provided no explanation of how face-to-face interviews were conducted safely during the pandemic. The process for distributing the online survey is also not described: authors should provide more details on this.

• The methods section does not describe any statistical analysis techniques used to analyze the data. This is a significant omission.

• There's no mention of statistical tests performed or p-values to indicate the significance of the findings.

• The results don't address potential confounding factors that could influence fear levels or vaccine confidence among healthcare workers.

• The discussion would benefit from comparing these findings to other relevant studies on healthcare worker vaccine confidence and fear levels during the pandemic.

• While regional differences are mentioned, there's minimal exploration of potential reasons behind these disparities. I would suggest expanding on this point.

• The conclusions could be strengthened with more specific recommendations for practice and policy based on the findings.

References:

1. https://doi.org/10.1038/s41398-020-01150-4

2. https://doi.org/10.1111/acps.13602

3. DOI: 10.3390/biomedicines12051083

**Do you want your identity to be public for this peer review?** For information about this choice, including consent withdrawal, please see our Privacy Policy

Reviewer #1: No

Reviewer #2: No

---

## [Author Response · Author response to Decision Letter 1]

9 Oct 2024

1. Please provide additional details regarding participant consent. In the ethics statement in the Methods and online submission information, please ensure that you have specified (1) whether consent was informed and what type you obtained (for instance, written or verbal, and if verbal, how it was documented and witnessed).

Response: We added the following details to the Ethical Considerations and Approval section of the Method. All participants were informed that their participation was voluntary and informed written consent was obtained by answering the first question before starting the survey (“to agree” or “not to agree”) to participate in the study.

2. If your study included minors, state whether you obtained consent from parents or guardians. If the need for consent was waived by the ethics committee, please include this information.

Thank you for raising this issue. In this study, we included adults aged 18 years or above. In this research, we included healthcare workers to assess their fear of new variants of COVID-19 and their confidence in the received vaccine.

3. If you are reporting a retrospective study of medical records or archived samples, please ensure that you have discussed whether all data were fully anonymized before you accessed them and/or whether the IRB or ethics committee waived the requirement for informed consent.

Response: Not applicable.

4. If patients provided informed written consent to have data from their medical records used in research, please include this information.

Response: Non-applicable

5. Once you have amended this/these statement(s) in the Methods section of the manuscript, please add the same text to the “Ethics Statement” field of the submission form (via “Edit Submission”). Done as requested

6. Please include a complete copy of PLOS’ questionnaire on inclusivity in global research in your revised manuscript.

Response: I have attached a copy of PLOS’ questionnaire on inclusivity in global research

7. Please upload a completed version of your questionnaire as Supporting Information when you resubmit your manuscript.

Response: A copy of the questionnaire was uploaded as a supplementary file.

8. Providing interested researchers with a durable point of contact ensures data will be accessible even if an author changes email addresses, institutions, or becomes unavailable to answer requests. Before we proceed with your manuscript, please also provide non-author contact information (phone/email/hyperlink) for a data access committee, ethics committee, or other institutional body to which data requests may be sent. If no institutional body is available to respond to requests for your minimal data, please consider if there any institutional representatives who did not collaborate in the study, and are not listed as authors on the manuscript, who would be able to hold the data and respond to external requests for data access? If so, please provide their contact information (i.e., email address). Please also provide details on how you will ensure persistent or long-term data storage and availability.

Response: We uploaded the data for repository and will be available for anyone.

https://osf.io/kmnsc/files/osfstorage?view_only=

9. PLOS requires an ORCID iD for the corresponding author in Editorial Manager on papers submitted after December 6th, 2016. Please ensure that you have an ORCID iD and that it is validated in the Editorial Manager

Response: I have attached my ORCID

10. Please amend your list of authors on the manuscript to ensure that each author is linked to an affiliation. Authors’ affiliations should reflect the institution where the work was done (if authors moved subsequently, you can also list the new affiliation stating “current affiliation:….” as necessary).

Response: Checked

11. Your ethics statement should only appear in the Methods section of your manuscript. If your ethics statement is written in any section besides the Methods, please delete it from any other sections. Response: Done as requested

---

## [Decision Letter · Decision Letter 1]

19 Nov 2024

Dear Dr. Ghazy,

Thank you for submitting your manuscript to PLOS ONE. After careful consideration, we feel that it has merit but does not fully meet PLOS ONE’s publication criteria as it currently stands. Therefore, we invite you to submit a revised version of the manuscript that addresses the points raised during the review process.

We look forward to receiving your revised manuscript.

Kind regards,

Stephen Dajaan Dubik, BSc, MPH, MPhil

Academic Editor

PLOS ONE

Journal Requirements:

Reviewers' comments:

Reviewer's Responses to Questions

**Comments to the Author**

Reviewer #2: All comments have been addressed

2. Is the manuscript technically sound, and do the data support the conclusions?

Reviewer #2: Yes

3. Has the statistical analysis been performed appropriately and rigorously?

Reviewer #2: Yes

4. Have the authors made all data underlying the findings in their manuscript fully available?

Reviewer #2: Yes

5. Is the manuscript presented in an intelligible fashion and written in standard English?

Reviewer #2: Yes

Reviewer #2: Dear Authors,

Thank you for the opportunity to review your manuscript. I appreciate the thorough and detailed responses to the comments I have raised in the precedent revision. It is evident that you have carefully considered the feedback and made significant improvements to the manuscript.

Below, I provide a summary of the changes and my responses to each of the comments.

• The study claims broad representation across various regions; however, clarifying the sampling method (e.g., were the samples proportionate to the healthcare workforce distribution?) would strengthen claims of cross-national comparison. Additionally, authors should specify why certain countries or regions may have lower representation could provide context for the sampling limitations.

• Since the authors report a correlation between mental health issues and vaccine hesitancy, additional context on how mental health was assessed or any screening tools used might be useful.

• The discussion might benefit from a more cautious interpretation of causality in observed relationships (e.g., family size, personal COVID-19 history) with vaccine confidence and fear. Since this is a cross-sectional study, it might be helpful to remind readers that causation cannot be firmly established.

• Although some limitations are implied, I recommend adding a dedicated limitations section discussing potential biases (e.g., self-reported data, convenience sampling) and limitations in the survey's cross-sectional design for a complete discussion.

In conclusion, the revisions made to the manuscript have significantly strengthened the quality and rigor of the work.

I commend the thoroughness of your responses and the attention to detail in addressing the concerns raised. The manuscript is now well-positioned for potential publication.

Sincerely,

Reviewer

**Do you want your identity to be public for this peer review?** For information about this choice, including consent withdrawal, please see our Privacy Policy

Reviewer #2: No

---

## [Author Response · Author response to Decision Letter 2]

8 Dec 2024

Reviewers' comments:

• Have the authors made all data underlying the findings in their manuscript fully available?

Response: We thank the reviewer for their comment. The data underlying our findings are fully available and comply with the PLOS Data Policy. Specifically, the following measures have been taken to ensure transparency and accessibility:

• We have included a detailed statement in the manuscript specifying that all data underlying the findings are available upon request by emailing the first author.

• The data points behind the means, medians, and variance measures mentioned in the manuscript have been preserved and are available for sharing. These data include the raw responses to the survey, stratified summary statistics, and supplementary analyses.

• Given the sensitive nature of the data and the need to protect participants' privacy, direct access to raw data is provided upon request. This approach aligns with ethical guidelines approved by the Institutional Review Board of the Faculty of Medicine, Alexandria University, Egypt (IRB number: 00012098).

If there are additional specifications or repositories preferred by the journal, we are willing to comply and upload the dataset accordingly.

• The study claims broad representation across various regions; however, clarifying the sampling method (e.g., were the samples proportionate to the healthcare workforce distribution?) would strengthen claims of cross-national comparison. Additionally, authors should specify why certain countries or regions may have lower representation could provide context for the sampling limitations.

Response: We thank the reviewer for this important observation. We acknowledge the need to clarify the sampling method and provide additional context regarding the representation of healthcare workers (HCWs) across regions. Below is our response:

• The study employed a convenience and snowball sampling approach to recruit HCWs from different countries. Collaborators from the Global Researcher Club (GRC), an international, voluntary, and non-profit research network, were tasked with recruiting participants from their respective regions. The sampling was not proportionate to the actual healthcare workforce distribution across countries or regions due to the limitations of available collaborators and logistical constraints.

• Although we aimed for broad representation, we acknowledge that the sample sizes from some regions (e.g., Western countries and certain parts of Eastern Asia and Latin America) are smaller compared to others like the Eastern Mediterranean and African regions. This disparity may reflect differences in the ease of access to HCWs, variations in collaborator networks, and the differing response rates among participants from these regions.

• We recognize that these sampling variations could introduce biases and limit the generalizability of our findings. However, we believe that the observed trends and associations remain valuable for highlighting regional disparities and guiding targeted interventions.

• We recommend that future studies consider employing proportionate sampling strategies or quota sampling methods based on the actual distribution of HCWs across regions. Additionally, using global health workforce statistics to guide recruitment efforts could improve representativeness.

We will include these clarifications in the revised manuscript under the "Methods" and "Limitations" sections to provide transparency and contextualize our findings.

• Since the authors report a correlation between mental health issues and vaccine hesitancy, additional context on how mental health was assessed or any screening tools used might be useful.

Response: We appreciate the reviewer’s valuable suggestion to provide additional context regarding the assessment of mental health issues, given its correlation with vaccine hesitancy in our study.

• Mental health was assessed in the study through self-reported responses to specific questions about the presence of mental health conditions, including stress, anxiety, sleep disorders, and obsessive-compulsive disorder (OCD). Participants were also asked if they had been diagnosed with schizophrenia or other psychiatric conditions. While these questions were based on participants' acknowledgment of existing conditions, either through personal awareness or prior diagnosis by a healthcare provider, they were not assessed using standardized tools.

• We recognize that the absence of validated screening tools, such as the Generalized Anxiety Disorder-7 (GAD-7) or Patient Health Questionnaire-9 (PHQ-9), limits the precision of our mental health measurements. We will include this limitation in the manuscript and emphasize the need for future studies to use validated instruments to enhance the robustness of findings related to mental health and vaccine hesitancy.

• We have added clarifications in the "Data Collection Tools" and "Discussion" sections of the manuscript, detailing how mental health issues were assessed and highlighting the limitations of the self-reported approach. These revisions aim to provide greater transparency and address the reviewer’s concern comprehensively.

Thank you for this insightful comment, which has strengthened our discussion of this important topic.

• The discussion might benefit from a more cautious interpretation of causality in observed relationships (e.g., family size, personal COVID-19 history) with vaccine confidence and fear. Since this is a cross-sectional study, it might be helpful to remind readers that causation cannot be firmly established.

Response: We fully acknowledge that the cross-sectional design of our study limits the ability to establish causation between variables such as family size or personal COVID-19 history and outcomes like vaccine confidence and fear. While we observed statistically significant associations, these should be interpreted as correlations rather than definitive causal relationships.

• To address this concern, we have revised the "Discussion" section to emphasize the observational nature of our findings. We have added the following clarification:

"It is important to note that the cross-sectional design of this study precludes any definitive conclusions about causality. The associations observed between variables such as family size or personal COVID-19 history and vaccine confidence or fear reflect correlations rather than causal relationships. Future longitudinal studies are necessary to explore these dynamics and determine causal pathways."

• This revision ensures that readers are reminded of the inherent limitations of the study design when interpreting the findings. Additionally, we will underscore the need for future studies employing longitudinal or experimental designs to further investigate these relationships.

Thank you for this valuable suggestion, which has enhanced the clarity and rigor of our discussion

• Although some limitations are implied, I recommend adding a dedicated limitations section discussing potential biases (e.g., self-reported data, convenience sampling) and limitations in the survey's cross-sectional design for a complete discussion.

Response: Done

---

## [Editor Report · Decision Letter 2]

25 Feb 2025

Dear Dr. Ghazy,

Thank you for submitting your manuscript to PLOS ONE. After careful consideration, we feel that it has merit but does not fully meet PLOS ONE’s publication criteria as it currently stands. Therefore, we invite you to submit a revised version of the manuscript that addresses the points raised during the review process.

We look forward to receiving your revised manuscript.

Kind regards,

Jennifer Tucker, PhD (Associate Editor, PLOS One) on behalf of

Stephen Dajaan Dubik, BSc, MPH, MPhil

Academic Editor

PLOS ONE
---

## [Author Response · Author response to Decision Letter 3]

6 Mar 2025

Dear Professor Stephen

Thank you for giving us the opportunity to enhance the quality of this manuscript. We have carefully addressed typos, grammatical errors, and redundancies to ensure it meets the high standards of your prestigious journal.

Sincerely,

Ramy Mohamed Ghazy

---

## [Decision Letter · Decision Letter 3]

4 Apr 2025

Dear Dr. Ghazy,

Thank you for submitting your manuscript to PLOS ONE. After careful consideration, we feel that it has merit but does not fully meet PLOS ONE’s publication criteria as it currently stands. Therefore, we invite you to submit a revised version of the manuscript that addresses the points raised during the review process.

We look forward to receiving your revised manuscript.

Kind regards,

Mona Gamal Mohamed

Academic Editor

PLOS ONE

Additional Editor Comments:

Dear Authors,

Thank you for your submission. Your study explores an important topic, but several key areas require revision to improve clarity, methodological rigor, and overall impact. Below are my detailed comments:

Title Revision: Please revise the title to include “…amid Emerging Variants” to better reflect the evolving pandemic context.

Introduction Restructuring: The introduction is too long and lacks coherence. Some sections include information that is not directly relevant to the study. Please restructure this section to ensure a logical flow and conciseness.

Sample Selection Bias: The study relies on a convenience sample of HCWs with internet access, which may introduce selection bias, particularly in regions with limited connectivity. Please discuss this limitation and explain any measures taken to mitigate its impact.

Data Collection Tools: This section should be more concise. Move detailed survey questions to an appendix and minimize descriptions of previously validated scales, referring instead to original sources.

Questionnaire Language: Clarify whether the questionnaire was administered in English across all countries. If so, discuss potential limitations for non-native speakers.

Results Section Revision: Given the large number of tables and figures, consider moving sociodemographic characteristics to a supplementary file. Additionally, refine the narrative to avoid excessive repetition of data already presented in the tables.

Sample Representativeness: The rationale for selecting specific countries is unclear. Please clarify why these countries were chosen and assess whether the sample represents the broader HCW population in each country based on demographics such as gender, age, and occupation.

Multivariate Analysis: The analysis does not appear to adjust for country-level differences, which may be a confounding factor. Given the diversity of the countries studied, it is crucial to account for these variations in attitudes toward vaccination. Please consider adjusting for country in the analysis.

Discussion Section Restructuring: The discussion is lengthy and difficult to follow. Structuring it into clear subsections will improve readability and enhance the clarity of your key findings.

Additionally, there is an interesting contrast between your findings and existing literature on vaccine hesitancy in the general population. Studies such as Dumitra et al. identify vaccine refusal primarily due to perceived lack of necessity or belief in natural immunity, while your study suggests that prior COVID-19 infection and vaccine side effects were associated with lower vaccine confidence among HCWs. This difference may be due to professional exposure, firsthand experience with COVID-19 severity, and evolving attitudes across different pandemic waves. Furthermore, studies like Savulescu et al. suggest that declining disease severity and the relatively short-lived protection of boosters may have influenced HCWs’ vaccine confidence. Incorporating this perspective into the discussion will strengthen your argument.

Thank you

Reviewers' comments:

Reviewer's Responses to Questions

**Comments to the Author**

Reviewer #3: All comments have been addressed

Reviewer #4: (No Response)

2. Is the manuscript technically sound, and do the data support the conclusions?

Reviewer #3: Yes

Reviewer #4: Partly

3. Has the statistical analysis been performed appropriately and rigorously?

Reviewer #3: Yes

Reviewer #4: Yes

4. Have the authors made all data underlying the findings in their manuscript fully available?

Reviewer #3: Yes

Reviewer #4: No

5. Is the manuscript presented in an intelligible fashion and written in standard English?

Reviewer #3: Yes

Reviewer #4: No

Reviewer #3: Dear Colleagues, thank you for your significant effort to collect and analyse such amount of data.

Several studies addressing vaccine hesitancy or acceptance in lay-people are documenting a more complex picture than just fear/extreme fear. A recent paper [Dumitra GG et al - Segmenting attitudes toward vaccination – behavioural insights into influenza vaccination refusal in Romania. GERMS. 2024;14(4):362-374. doi: 10.18683/germs.2024.1446] documented 3 distinct latent-class groups of people regarding vaccine and disease perception. Highest refusal was present in people that perceived lack of necessity and belief in natural immunity. A smaller degree of refusal was documented in people with past negative experiences with vaccines or distrust in vaccines.

Can you comment about these findings compared with your statement in page 25 "personal history of COVID-19 infection, and experiencing side effects of COVID vaccination were associated with lower vaccine confidence" after stating in page 24 "Research investigating vaccine confidence in HCWs with a previous history of COVID-19 infection also yields mixed results, with some studies reporting high confidence [75], while others suggest low confidence [76]. It is worth emphasizing that the firsthand experience of the disease's severity and its impact play a crucial role in shaping confidence toward vaccination"

These seem to document different perception models from HCW in your study and general population, probably because of major differences in disease perception [did you ask in your study if responding HCW were working in a unit where COVID-19 patients were isolated/treated?].

Experiencing first-hand disease severity decrease during different waves of pandemic (after Delta wave all subsequent Omicron lineages were less and less devastating), several groups of HCW decreased in fear and potentially developed a more relaxed behaviour and less adoption of recurrent boosters. Some sense of "futility" of boosting was inflicted by a certain perception bias, when confronted with the relative short period of protection gained by boosting as documented in an European study performed in 10 countries and 19 hospitals (Savulescu C, et al. Incidence of SARS-CoV-2 Infection Among European Healthcare Workers and Effectiveness of the First Booster COVID-19 Vaccine, VEBIS HCW Observational Cohort Study, May 2021–May 2023. Vaccines. 2024; 12(11):1295. https://doi.org/10.3390/vaccines12111295).

Reviewer #4: Dear Authors,

I want to congratulate you on the topic of your article, given the crucial importance of vaccination among healthcare professionals from a preventive perspective. Understanding the psychosocial factors influencing COVID-19 vaccination is essential—not only for protecting healthcare workers themselves but also for their patients, as they play a direct role in their immunization efforts.

However, I believe the manuscript requires some revisions before it can be considered for publication. Below, I outline the necessary changes and provide several questions that I would like to see addressed:

1. Title: Please revise to include the phrase “…amid Emerging Variants”.

2. Introduction: The section is too long and needs to be restructured. It includes a significant amount of information that is not directly relevant to the study. Additionally, there is a lack of coherence between successive paragraphs, which seem to have been written independently. The introduction should have a clear and logical flow, which is currently missing. Please revise.

3. Sample Selection Bias: The study relies on a convenience sample, including only healthcare professionals with internet access. Could this introduce a significant selection bias, particularly in countries where internet access may be limited? How was this issue addressed? Was any measure taken to mitigate its impact?

4. Data Collection Tools section: This section should be more concise, with detailed information on survey questions moved to an appendix. Additionally, since previously validated scales were used, their descriptions should be minimized, referring readers to the original publications for further details.

5. Language of the Questionnaire: was the questionnaire administered in English in all countries? If so, could this be a limitation for participants in certain regions?

6. Results Section: Given the large number of tables and figures, I suggest moving the initial tables on sociodemographic characteristics to a supplementary file. Moreover, the narrative description is overly detailed, often repeating information already provided in the tables. Please revise.

7. Sample Representativeness: While the study appears methodologically well performed, I have concerns about the representativeness of the sample. First, the rationale for grouping these specific countries together is unclear. Second, to what extent do the selected healthcare professionals represent the broader population of healthcare workers in each country? Was any assessment of representativeness conducted? For example, does the sample reflect national distributions in terms of gender, age group, and occupation?

8. Multivariate Analysis: The analysis does not appear to adjust for country. Given the diversity of the countries studied, do you not consider this a potential confounding factor? Beliefs and attitudes toward vaccination are likely influenced by the socioeconomic and cultural contexts of each country, making it essential to account for these differences in the analysis. Please consider this suggestion.

9. Discussion Section: The discussion lacks structure and is difficult to follow. Given its length, I recommend restructuring it into subsections to improve readability and comprehension.

**Do you want your identity to be public for this peer review?** For information about this choice, including consent withdrawal, please see our Privacy Policy

Reviewer #3: **Yes: ** Mihai Craiu

Reviewer #4: No

---

## [Author Response · Author response to Decision Letter 4]

20 Apr 2025

Dear Editorial Office,

Thank you for reviewing the manuscript I submitted for PLOS ONE entitled (Cross-National Disparities in Healthcare Workers' Perceptions: Examining Fear of Infection and Confidence in COVID-19 Vaccines amid Emerging Variants). Your efforts in reviewing my manuscript are appreciated. Your comments have been instrumental in enhancing the quality of our research, and we have carefully considered them. We are pleased to present the revised version of our manuscript, incorporating the changes you suggested

We would like to inform you that our manuscript was initially accepted and sent to production. However, we made minor edits afterward to enhance the quality of our work. As a result, the manuscript underwent another round of review and received a minor revision decision. The editor apologized for this additional review round, which occurred solely because of our efforts to improve the manuscript’s quality

As regards your valuable comments on the following:

Editor Comments:

1- Title Revision: Please revise the title to include “…amid Emerging Variants” to better reflect the evolving pandemic context.

Author response: Thank you for your valuable suggestion. We agree that incorporating “amid Emerging Variants” enhances the relevance of the title within the evolving context of the COVID-19 pandemic. Accordingly, the revised title is: “Cross-National Disparities in Healthcare Workers' Perceptions: Examining Fear of Infection and Confidence in COVID-19 Vaccines amid Emerging Variants.”

2- Introduction Restructuring: The introduction is too long and lacks coherence. Some sections include information that is not directly relevant to the study. Please restructure this section to ensure a logical flow and conciseness.

Author response: Thank you for your valuable suggestion. It has been done as requested

3- Sample Selection Bias: The study relies on a convenience sample of HCWs with internet access, which may introduce selection bias, particularly in regions with limited connectivity. Please discuss this limitation and explain any measures taken to mitigate its impact.

Author response: Thank you for your valuable comments. The data was collected through an online anonymous survey and face-to-face distribution of hard copies of surveys to target participants during working hours. The online survey was uploaded to Google Forms, then distributed through social media platforms (Facebook and Twitter) and sent via email, WhatsApp, and Telegram. The printed hard copy of the study questionnaire was distributed to targeted HCWs to complete it to overcome any challenges with internet access.

Also, we recommended that future studies should consider employing proportionate or quota sampling strategies based on global health workforce distribution to improve representativeness.

4- Data Collection Tools: This section should be more concise. Move detailed survey questions to an appendix and minimize descriptions of previously validated scales, referring instead to original sources.

Author response: Thank you for your valuable comments. It was done as requested and the details on the supplementary file and all related and required information remain in the main manuscript

5- Questionnaire Language: Clarify whether the questionnaire was administered in English across all countries. If so, discuss potential limitations for non-native speakers.

Author response: Thank you for your valuable comments. the questionnaire was used in both Arabic and English versions

6- Results Section Revision: Given the large number of tables and figures, consider moving sociodemographic characteristics to a supplementary file. Additionally, refine the narrative to avoid excessive repetition of data already presented in the tables.

Author response: Thank you for your suggestion. However, we believe it is important to keep the sociodemographic characteristics table in the main manuscript for the following reasons:

• It describes our sample in detail and demonstrates its balance across key demographics such as age, gender, and profession.

• These characteristics are directly related to the main outcomes (fear and vaccine confidence) and help the reader interpret our findings more clearly.

• We also wanted to ensure transparency, especially in a multi-country study, where understanding the sample makeup is essential.

Regarding the narrative, we were careful to highlight only the most important findings and avoid repeating data already shown in the tables.

7- Sample Representativeness: The rationale for selecting specific countries is unclear. Please clarify why these countries were chosen and assess whether the sample represents the broader HCW population in each country based on demographics such as gender, age, and occupation.

Author response: Thank you for this insightful comment. We appreciate the opportunity to clarify the rationale for our country selection and address the representativeness of our sample.

The countries included in the study were selected based on the feasibility of data collection through our existing network of collaborators in the Global Researcher Club (GRC), an international, voluntary, and non-profit scientific research network. Our aim was to achieve a diverse sample across multiple WHO regions, including the Eastern Mediterranean Region (EMR), African Region (AFR), Western Region, and other global areas such as Asia and Latin America. Collaborators from these countries had prior experience with survey-based research and were able to facilitate recruitment and translation where necessary.

We acknowledge that this convenience and snowball sampling approach may limit the representativeness of the sample. To address this concern, we have now added a section to the Discussion highlighting this limitation and recommending that future studies use quota or stratified sampling to ensure representativeness in terms of gender, age, and professional categories across countries.

8- Multivariate Analysis: The analysis does not appear to adjust for country-level differences, which may be a confounding factor. Given the diversity of the countries studied, it is crucial to account for these variations in attitudes toward vaccination. Please consider adjusting for country in the analysis.

Author response: Thank you for this important point. We agree that cultural and socioeconomic differences across countries may influence fear and vaccine confidence. However, our goal was to explore generalizable trends across regions, and our multivariate analysis adjusted for world region (e.g., EMR, AFR, Western, Other), which we used as a proxy for broader contextual differences.

While adjusting for individual countries was limited by unequal sample sizes across countries, this may have reduced model stability or introduced bias. We acknowledge this as a limitation and have added a statement in the Discussion section recommending that future studies use country-level adjustment or stratified analyses where sample sizes allow.

9- Discussion Section Restructuring: The discussion is lengthy and difficult to follow. Structuring it into clear subsections will improve readability and enhance the clarity of your key findings.

Author response: Thank you for your valuable feedback. We appreciate your suggestion regarding the structure of the Discussion section. In response, we have carefully revised and reorganized the section into clearly defined subsections, each addressing a specific aspect of our findings.

Additionally, there is an interesting contrast between your findings and existing literature on vaccine hesitancy in the general population. Studies such as Dumitra et al. identify vaccine refusal primarily due to perceived lack of necessity or belief in natural immunity, while your study suggests that prior COVID-19 infection and vaccine side effects were associated with lower vaccine confidence among HCWs. This difference may be due to professional exposure, firsthand experience with COVID-19 severity, and evolving attitudes across different pandemic waves. Furthermore, studies like Savulescu et al. suggest that declining disease severity and the relatively short-lived protection of boosters may have influenced HCWs’ vaccine confidence. Incorporating this perspective into the discussion will strengthen your argument.

Author response: Thank you for this thoughtful and insightful comment. We agree that contrasting our findings with existing literature on vaccine hesitancy in the general population adds important context to our study. We have now integrated this perspective into the revised Discussion section, specifically under the subsection “Interplay Between Fear and Vaccine Confidence.” We also referenced studies by Dumitra et al. and Savulescu et al. to highlight how the professional experiences of HCWs may shape their confidence in COVID-19 vaccines differently from the general population. This addition helps contextualize our findings within the broader literature and enhances the strength of our interpretation.

Reviewers' comments:

Reviewer #3:

1- Several studies addressing vaccine hesitancy or acceptance in lay-people are documenting a more complex picture than just fear/extreme fear. A recent paper [Dumitra GG et al - Segmenting attitudes toward vaccination – behavioural insights into influenza vaccination refusal in Romania. GERMS. 2024;14(4):362-374. doi: 10.18683/germs.2024.1446] documented 3 distinct latent-class groups of people regarding vaccine and disease perception. Highest refusal was present in people that perceived lack of necessity and belief in natural immunity. A smaller degree of refusal was documented in people with past negative experiences with vaccines or distrust in vaccines. Can you comment about these findings compared with your statement in page 25 "personal history of COVID-19 infection, and experiencing side effects of COVID vaccination were associated with lower vaccine confidence" after stating in page 24 "Research investigating vaccine confidence in HCWs with a previous history of COVID-19 infection also yields mixed results, with some studies reporting high confidence [75], while others suggest low confidence [76]. It is worth emphasizing that the firsthand experience of the disease's severity and its impact play a crucial role in shaping confidence toward vaccination"

Author response: Thank you for this insightful observation and for highlighting the study by Dumitra et al., which enriches the discussion on the heterogeneity of vaccine hesitancy. We fully agree that vaccine attitudes extend beyond simple fear and are shaped by multiple psychological, experiential, and cognitive dimensions.

To address this, we have now expanded the relevant section of the Discussion to better contrast our findings among HCWs with the behavioural segmentation observed in the general population. Specifically, we highlight that while Dumitra et al. identified a lack of perceived necessity and belief in natural immunity as key predictors of hesitancy in the lay public, our study among HCWs revealed that prior COVID-19 infection and negative post-vaccination experiences were associated with decreased confidence. This contrast likely reflects differences in baseline health literacy, exposure to clinical realities, and risk assessment frameworks.

2- These seem to document different perception models from HCW in your study and general population, probably because of major differences in disease perception [did you ask in your study if responding HCW were working in a unit where COVID-19 patients were isolated/treated?].

Author response: Thank you for this thoughtful comment. You are correct in noting that disease perception models likely differ between HCWs and the general population due to differences in risk exposure and professional experience. While our survey captured detailed sociodemographic and health information, including history of COVID-19 infection and vaccine side effects—it did not specifically ask whether HCWS were assigned to COVID-19 treatment or isolation units. We acknowledge this as a limitation in the study and have added a statement to the Discussion section to reflect this. We also suggest that future research incorporate clinical role or unit assignment as a potential modifier of vaccine confidence and fear.

3- Experiencing first-hand disease severity decrease during different waves of pandemic (after Delta wave all subsequent Omicron lineages were less and less devastating), several groups of HCW decreased in fear and potentially developed a more relaxed behaviour and less adoption of recurrent boosters. Some sense of "futility" of boosting was inflicted by a certain perception bias, when confronted with the relative short period of protection gained by boosting as documented in an European study performed in 10 countries and 19 hospitals (Savulescu C, et al. Incidence of SARS-CoV-2 Infection Among European Healthcare Workers and Effectiveness of the First Booster COVID-19 Vaccine, VEBIS HCW Observational Cohort Study, May 2021–May 2023. Vaccines. 2024; 12(11):1295. https://doi.org/10.3390/vaccines12111295).

Author response: We appreciate the reviewer’s insightful comment highlighting the evolving dynamics of disease severity perception among HCWs. Indeed, as the pandemic progressed and Omicron lineages replaced more severe variants like Delta, HCWs may have adjusted their risk assessments, leading to a decline in fear and a perceived diminishing value in repeated booster vaccinations. We have now incorporated this point into the Discussion section and cited the European VEBIS HCW Cohort Study by Savulescu et al. (2024), which supports the observation of declining perceived utility of boosters due to their short-lived protective effect.

Reviewer #4:

1- Title: Please revise to include the phrase “…amid Emerging Variants”

Author response: Thank you for your valuable suggestion. We agree that incorporating “amid Emerging Variants” enhances the relevance of the title within the evolving context of the COVID-19 pandemic. Accordingly, the revised title is: “Cross-National Disparities in Healthcare Workers' Perceptions: Examining Fear of Infection and Confidence in COVID-19 Vaccines amid Emerging Variants.”

2- Introduction: The section is too long and needs to be restructured. It includes a significant amount of information that is not directly relevant to the study. Additionally, there is a lack of coherence between successive paragraphs, which seem to have been written independently. The introduction should have a clear and logical flow, which is currently missing. Please revise.

Author response: Thank you for this insight. It has been reconstructed for better coherence.

3- Sample Selection Bias: The study relies on a convenience sample, including only healthcare professionals with internet access. Could this introduce a significant selection bias, particularly in countries where internet access may be limited? How was this issue addressed? Was any measure taken to mitigate its impact

Author response: Thank you for your valuable comments. The data was collected through an online anonymous survey and face-to-face distribution of hard copies of surveys to target participants during working hours. The online survey was uploaded to Google Forms, then distributed through social media platforms (Facebook and Twitter) and sent via email, WhatsApp, and Telegram. The printed hard copy of the study questionnaire was distributed to targeted HCWs to complete it to overcome any challenges with internet access.

Also, we recommended that future studies should consider employing proportionate or quota sampling strategies based on global health workforce distribution to improve representativeness.

4- Data Collection Tools section: This section should be more concise, with detailed information on survey questions moved to an appendix. Additionally, since previously validated scales were used, their descriptions should be minimized, referring readers to the original publications for further details.

Author response: Thank you for your valuable comments. It was done as requested and the details on the supplementary file and all related and required information remain in the main manuscript

Additionall

---

## [Decision Letter · Decision Letter 4]

10 Jul 2025

Dear Dr. Ghazy,

Thank you for submitting your manuscript to PLOS ONE. After careful consideration, we feel that it has merit but does not fully meet PLOS ONE’s publication criteria as it currently stands. Therefore, we invite you to submit a revised version of the manuscript that addresses the points raised during the review process.

Thank you for your submission addressing a timely and important topic. The focus on vaccine hesitancy during the pandemic is highly relevant, and your efforts to capture cross-national perspectives are commendable. However, to improve the scientific rigor and clarity of the manuscript, several key areas require attention. Please ensure that all figures are correctly embedded in the manuscript and that abbreviations such as "VH" are clearly defined upon first use. Additionally, clarify ambiguous response categories (e.g., "others" in place of residence) to enhance interpretability. The Results section could be more concise by removing redundant tables (e.g., Tables 1 and 2), as the content is already described in the narrative. It would also be helpful to specify the type of vaccine (e.g., monovalent or bivalent) referenced, to align with evolving public health contexts. While the cross-sectional design limits causal inference, your findings still offer valuable insights and may serve as a useful foundation for future research and policy discussions. I encourage you to revise the manuscript accordingly to improve its clarity, coherence, and contribution to the field.

We look forward to receiving your revised manuscript.

Kind regards,

Mona Gamal Mohamed

Academic Editor

PLOS ONE

Journal Requirements:

Additional Editor Comments:

The aim of this research is timely and relevant, particularly in the context of ongoing global discussions surrounding vaccine hesitancy and the evolving dynamics of the COVID-19 pandemic. While the cross-sectional design and use of a simple questionnaire limit the depth of scientific inquiry, the inclusion of participants from multiple countries adds value by offering broad, generalized insights that could inform future research and public health policy. However, several critical revisions are necessary to strengthen the manuscript. First, clarify abbreviations such as "VH" upon first use and specify what is included under "others" in the place of residence category. Please also ensure that all figures are correctly displayed and embedded in the manuscript. Consider whether it would be beneficial to indicate the version of the vaccine (monovalent or bivalent) to align with variant-specific developments. To improve the clarity and conciseness of the Results section, we recommend removing Tables 1 and 2, as their content is already described in the text, and consider including Supplementary Table 2, which appears to be missing. Addressing these points will enhance the manuscript’s overall clarity, relevance, and impact for a broad audience.

Reviewers' comments:

Reviewer's Responses to Questions

**Comments to the Author**

Reviewer #3: All comments have been addressed

Reviewer #5: (No Response)

Reviewer #6: (No Response)

2. Is the manuscript technically sound, and do the data support the conclusions?

Reviewer #3: Yes

Reviewer #5: Yes

Reviewer #6: Partly

3. Has the statistical analysis been performed appropriately and rigorously?

Reviewer #3: Yes

Reviewer #5: N/A

Reviewer #6: Yes

4. Have the authors made all data underlying the findings in their manuscript fully available?

Reviewer #3: Yes

Reviewer #5: Yes

Reviewer #6: No

5. Is the manuscript presented in an intelligible fashion and written in standard English?

Reviewer #3: Yes

Reviewer #5: Yes

Reviewer #6: No

Reviewer #3: Dear colleagues, thank you for the persistent effort made in improvement of your research especially in the Discussion section that was significantly improved by inserting clarifications in sections like "Fear of Emerging COVID-19 Variants Among HCWs", "Confidence in the received COVID-19 Vaccines" or "Interplay between fear of new SARS-Cov-2 variant infection and confidence in the received vaccine". Also valuable improvement has been implemented in Study Limitations section. Congratulations for your detailed revision of this paper.

Reviewer #5: Thanks to all authors for their competitive work, please find some comments below:

1- Introduction: A study on 2,336 HWCs reported an increasing rate of severe COVID-19 fear from 9% to 15% please give details how you concluded its increase, is it a longitudinal study?

2- Methods: You mentioned (A nationwide); however, this study has a global outreach, what nation did you mean?

3- Regarding age distribution, 86 was the maximum, this age for healthcare worker is a bit weird.

4- In conclusion, HCWs exhibited notable fear of SARS-CoV-2 new variants with low confidence. Better to add infection to be notable fear of infection with SARS-CoV-2 new variants with low confidence

5- Sample size calculation: We multiplied by 4 to compensate for stratification (Western region, Arab region, African region, and Others). I would advise you to add the total sample size

6- Kindly note that WHO classifies variants as variants of concern, variants of interest and variants under monitoring. CDC classifies variants as variants of concern, variants of interest, and variants of high consequence. Nomenclature by WHO is a bit more recent.

7- Do you add the version of vaccine Monovalent / bivalent to coincide with the updated form of variants?

8- Please clarify what is VH in the first time its mentioned in the text.

9- Please take care that figures are not displayed in the manuscript provided to me for review, you must ensure their inclusion.

10- Place of residence included others, please define what are the others?

11- Discussion: Key brain regions, such as the amygdala, insula, and prefrontal cortex, play important roles in processing fear, risk perception, and decision making. What does this statement add to the discussion?

Reviewer #6: The aim of this research is pertinent topic for examination at this phase of pandemic and with the common concerns about the pharmaceutical intervention. The cross-sectional study design, with broad questions in simple questionnaire survey, mean that depth is lacking for true scientific inquiry (thus, "partly" for Review Question #2), but the inclusion of multiple countries mean that these generalized findings may help understanding for non-specific audience of readers. The claims could be useful, even if refuted by more rigorous design, for further medical research and informing public health policy.

With these strengths, there are some suggestions (in response to Question #1) made by previous reviewers which could vastly improve the study manuscript. I agree that the Results sections has too many tables with the same information are already written in the text. My suggestion is to delete Tables 1 and 2 to reduce the size of this section and possibly include the Supplementary Table #2 (unavailable in the current manuscript).

Major omissions in document will require fix for the final submission (in response to Question #4): Figure 1 & 2 are missing from Pages 24 & 27 in the version I received; and there is no "Supplementary Table 2" available. I was unable to review these inserts of results.

As noted in previous reviews and still not resolved satisfyingly, the wording in the manuscript is imprecise and also the flow of introduction & discussion are too general and off-topic ("no" on Question #5) seemingly like an early draft. I have some specific improvements here:

Page 1: Short title: Misspelling "SARS-Cov2"

Page 1: Abstract conclusion: Revise to "In conclusion, HCWs exhibited notable fear of emerging SARS-CoV2 variants and there is low level of confidence in COVID-19 vaccination"

Page 16: In stead of "vaccine effectiveness concerns surfaced", there were new questions about "duration of protection" and "effectiveness against new variants" questions post-Omicron variant emergence

"SARS-Cov-2 vaccine approval" should be written "COVID-19 vaccine approval,".. Moreover, there are concerns over "how immunity from vaccines might influence viral evolution of spike protein"

Page 18: Use "of either sex" because you can't have a study participant who is both sexes at once

Page 23: Hypertension is chronic "condition" not disease. Use "the most reported health problems"

Page 28: Table 5 Row title use "Area of practice in healthcare work" deleting Field of study to support statement in discussion on Page 32 "Working as a physician, dentist, pharmacist, or physiotherapist was associated with

lower levels of fear."

Page 31 "Neural substrates" is off topic and should be removed. Focus discussion on study related measurements, like which HCW had increased fear, or Discussion documented number of deaths in surveyed countries correlate with fear?

Again, a general discussion about marital happiness is out of focus on COVID-19 fear. What about illness support through marriage

Page 32 Discussion missed a large issue of interest, on if Confidence is related according to vaccine received?

Page 34 What about high non-response rate? Is the missing population a limitation in your findings?

Page 34 Length of time (2 years +) since vaccines introduced should be discussed as aspect of study finding on confidence of vaccine (as well as fear of new variants); Furthermore, have better treatments become available during the time since pandemic to ease fears of death and severe illness from new COvID-19 infection

**Do you want your identity to be public for this peer review?** For information about this choice, including consent withdrawal, please see our Privacy Policy

Reviewer #3: **Yes: ** Mihai Craiu

Reviewer #5: No

Reviewer #6: **Yes: ** C. Jason McKnight

---

## [Author Response · Author response to Decision Letter 5]

13 Jul 2025

Dear Reviewer Team,

Thank you very much for your time and feedback. We go over all the comments and provide our responses.

Additional Editor Comments:

The aim of this research is timely and relevant, particularly in the context of ongoing global discussions surrounding vaccine hesitancy and the evolving dynamics of the COVID-19 pandemic. While the cross-sectional design and use of a simple questionnaire limit the depth of scientific inquiry, the inclusion of participants from multiple countries adds value by offering broad, generalized insights that could inform future research and public health policy. However, several critical revisions are necessary to strengthen the manuscript.

• First, clarify abbreviations such as "VH" upon first use and specify what is included under "others" in the place of residence category.

Response: Thank you for your comment. We have revised the manuscript to clarify what is included under the “others” category for place of residence. Specifically, “others” refers to healthcare workers who reported non-standard or unlisted living areas such as refugee camps, temporary housing, or remote settlements that do not fall under the predefined categories of urban/city, rural/village, or desert/mountain.

• Please also ensure that all figures are correctly displayed and embedded in the manuscript.

Response: Done

• Consider whether it would be beneficial to indicate the version of the vaccine (monovalent or bivalent) to align with variant-specific developments.

Response: Thank you for this insightful suggestion. Unfortunately, the survey did not collect information on the specific version of the COVID-19 vaccine (monovalent vs. bivalent) received by participants. At the time of data collection, most vaccinations administered in the participating regions were from the initial rollout phases, and bivalent vaccines were not yet widely available. Therefore, we were unable to differentiate vaccine versions in our analysis. We have added a note to the Limitations section to reflect this point.

• To improve the clarity and conciseness of the Results section, we recommend removing Tables 1 and 2, as their content is already described in the text, and consider including Supplementary Table 2, which appears to be missing. Addressing these points will enhance the manuscript’s overall clarity, relevance, and impact for a broad audience.

Response: Done

Reviewer #3:

Dear colleagues, thank you for the persistent effort made in improvement of your research especially in the Discussion section that was significantly improved by inserting clarifications in sections like "Fear of Emerging COVID-19 Variants Among HCWs", "Confidence in the received COVID-19 Vaccines" or "Interplay between fear of new SARS-Cov-2 variant infection and confidence in the received vaccine". Also, valuable improvements have been implemented in Study Limitations section. Congratulations on your detailed revision of this paper.

Response: Thank you very much for your feedback and advice

Reviewer #5:

Thanks to all authors for their competitive work, please find some comments below:

1- Introduction: A study on 2,336 HWCs reported an increasing rate of severe COVID-19 fear from 9% to 15% please give details how you concluded its increase, is it a longitudinal study?

Response:

We appreciate the reviewer’s thoughtful observation. The referenced study by Moretti et al. (2022) is indeed a prospective cohort study, conducted at a single center, which followed healthcare workers over time. The reported increase in severe COVID-19 fear from 9% to 15% reflects a temporal change within the same cohort, confirming that the study had a longitudinal design.

2- Methods: You mentioned (A nationwide); however, this study has a global outreach, what nation did you mean?

Response:

Thank you for highlighting this inconsistency. You are correct, the study involved participants from multiple countries and regions, and the term “nationwide” inaccurately implies it was restricted to a single country. We have revised the wording in the Methods section to reflect the global scope of the survey.

3- Regarding age distribution, 86 was the maximum, this age for healthcare worker is a bit weird.

Response:

Thank you for this insightful comment. We agree that 86 years may seem atypical for an actively practicing healthcare worker. However, our study included a broad definition of healthcare workers, encompassing not only clinical staff but also retired professionals still engaged in part-time roles or consultancy, academic staff, and volunteers involved in healthcare settings. In some countries, particularly where healthcare infrastructure is stretched, retired professionals or older individuals may continue contributing to the healthcare system in advisory, supervisory, or training capacities.

4- In conclusion, HCWs exhibited notable fear of SARS-CoV-2 new variants with low confidence. Better to add infection to be notable fear of infection with SARS-CoV-2 new variants with low confidence

Response: Done

5- Sample size calculation: We multiplied by 4 to compensate for stratification (Western region, Arab region, African region, and Others). I would advise you to add the total sample size

Response: Done

6- Kindly note that WHO classifies variants as variants of concern, variants of interest and variants under monitoring. CDC classifies variants as variants of concern, variants of interest, and variants of high consequence. Nomenclature by WHO is a bit more recent.

Response: In this study, we used the term “emerging variants” to refer broadly to newly identified or recently spreading SARS-CoV-2 variants, without assigning them to specific official classifications.

7- Do you add the version of vaccine Monovalent / bivalent to coincide with the updated form of variants?

Response: Thank you for this insightful suggestion. Unfortunately, the survey did not collect information on the specific version of the COVID-19 vaccine (monovalent vs. bivalent) received by participants. At the time of data collection, most vaccinations administered in the participating regions were from the initial rollout phases, and bivalent vaccines were not yet widely available. Therefore, we were unable to differentiate vaccine versions in our analysis. We have added a note to the Limitations section to reflect this point.

8- Please clarify what VH is the first time its mentioned in the text.

Response: Done

9- Please take care that figures are not displayed in the manuscript provided to me for review, you must ensure their inclusion.

Response: Done

10- Place of residence included others, please define what the others are?

Response: Thank you for your comment. We have revised the manuscript to clarify what is included under the “others” category for place of residence. Specifically, “others” refers to healthcare workers who reported non-standard or unlisted living areas such as refugee camps, temporary housing, or remote settlements that do not fall under the predefined categories of urban/city, rural/village, or desert/mountain.

11- Discussion: Key brain regions, such as the amygdala, insula, and prefrontal cortex, play important roles in processing fear, risk perception, and decision making. What does this statement add to the discussion?

Response:

Thank you for your insightful comment. We agree that the original statement required further contextualization. We have revised the paragraph to directly relate the neurocognitive basis of fear and risk perception to the observed extreme fear among HCWs. This addition helps to explain the psychological underpinnings behind fear responses to emerging variants and how such fear may influence behavior and attitudes toward vaccination.

Reviewer #6:

The aim of this research is pertinent topic for examination at this phase of pandemic and with common concerns about the pharmaceutical intervention. The cross-sectional study design, with broad questions in simple questionnaire survey, mean that depth is lacking for true scientific inquiry (thus, "partly" for Review Question #2), but the inclusion of multiple countries means that these generalized findings may help understanding for non-specific audience of readers. The claims could be useful, even if refuted by more rigorous design, for further medical research and informing public health policy.

With these strengths, there are some suggestions (in response to Question #1) made by previous reviewers which could vastly improve the study manuscript. I agree that the Results sections has too many tables with the same information already written in the text.

• My suggestion is to delete Tables 1 and 2 to reduce the size of this section and possibly include the Supplementary Table #2 (unavailable in the current manuscript).

Response: Done

• Major omissions in documents will require fix for the final submission (in response to Question #4): Figure 1 & 2 are missing from Pages 24 & 27 in the version I received; and there is no "Supplementary Table 2" available. I was unable to review these inserts of results.

Response: Done

As noted in previous reviews and still not resolved satisfyingly, the wording in the manuscript is imprecise and also the flow of introduction & discussion are too general and off-topic ("no" on Question #5) seemingly like an early draft. I have some specific improvements here:

• Page 1: Short title: Misspelling "SARS-Cov2"

Done

• Page 1: Abstract conclusion: Revise to "In conclusion, HCWs exhibited notable fear of emerging SARS-CoV2 variants and there is low level of confidence in COVID-19 vaccination"

Done

• Page 16: In stead of "vaccine effectiveness concerns surfaced", there were new questions about "duration of protection" and "effectiveness against new variants" questions post-Omicron variant emergence

Done

• "SARS-Cov-2 vaccine approval" should be written "COVID-19 vaccine approval,".. Moreover, there are concerns over "how immunity from vaccines might influence viral evolution of spike protein"

Done

• Page 18: Use "of either sex" because you can't have a study participant who is both sexes at once

Done

• Page 23: Hypertension is chronic "condition" not disease. Use "the most reported health problems"

Done

• Page 28: Table 5 Row title use "Area of practice in healthcare work" deleting Field of study to support statement in discussion on Page 32 "Working as a physician, dentist, pharmacist, or physiotherapist was associated with lower levels of fear."

Done

• Page 31 "Neural substrates" is off topic and should be removed. Focus discussion on study related measurements, like which HCW had increased fear, or Discussion documented number of deaths in surveyed countries correlate with fear?

Clarified

• Again, a general discussion about marital happiness is out of focus on COVID-19 fear. What about illness support through marriage

Response:

Thank you for this insightful comment. We agree that a general discussion of marital happiness may diverge from the core focus of our study. In response, we have revised the discussion to emphasize how marital relationships may offer emotional and practical support during illness, which could help mitigate fear associated with infectious disease outbreaks.

• Page 32 Discussion missed a large issue of interest, on if Confidence is related according to vaccine received?

Response:

Thank you for this important observation. We acknowledge that the discussion did not explicitly address the potential association between the type of vaccine received and healthcare workers’ confidence in COVID-19 vaccination. While our study collected data on vaccine types, our primary analysis did not stratify confidence levels by specific vaccine brands. This can be an area for further investigation.

• Page 34 What about high non-response rate? Is the missing population a limitation in your findings?

Response:

Thank you for raising this important point. We acknowledge that our study did not formally assess or report the response rate, as the survey was disseminated through online platforms using a snowball sampling technique, and the denominator of those who received the survey is unknown. Therefore, we could not estimate the non-response rate accurately.

• Page 34 Length of time (2 years +) since vaccines introduced should be discussed as aspect of study finding on confidence of vaccine (as well as fear of new variants); Furthermore, have better treatments become available during the time since pandemic to ease fears of death and severe illness from new COvID-19 infection

Done

---

## [Decision Letter · Decision Letter 5]

23 Oct 2025

Cross-National disparities in healthcare workers' perceptions: examining fear of infection and confidence in the received COVID-19 vaccines amid emerging variants

PONE-D-24-28390R5

Dear Dr. Ramy

We’re pleased to inform you that your manuscript has been judged scientifically suitable for publication and will be formally accepted for publication once it meets all outstanding technical requirements.

Kind regards,

Eman Abdelaziz Rashad Dabou, Ph.D, M.S.N

Academic Editor

PLOS ONE

Additional Editor Comments (optional):

After reviewing the peer reviewers’ comments and noting that all concerns satisfactorily addressed, I accept the manuscript.

Reviewers' comments:

Reviewer's Responses to Questions

**Comments to the Author**

Reviewer #3: All comments have been addressed

Reviewer #5: All comments have been addressed

Reviewer #6: All comments have been addressed

2. Is the manuscript technically sound, and do the data support the conclusions?

Reviewer #3: Yes

Reviewer #5: Yes

Reviewer #6: Yes

3. Has the statistical analysis been performed appropriately and rigorously?

Reviewer #3: Yes

Reviewer #5: I Don't Know

Reviewer #6: Yes

4. Have the authors made all data underlying the findings in their manuscript fully available?

Reviewer #3: Yes

Reviewer #5: Yes

Reviewer #6: Yes

5. Is the manuscript presented in an intelligible fashion and written in standard English?

Reviewer #3: Yes

Reviewer #5: Yes

Reviewer #6: Yes

Reviewer #3: Dear colleagues in page 33 of your revised version you are stating that tailoring approach is paramount: "These findings underscore the need for providing targeted educational programs and personalized psychological support and encourage open communication to address these concerns effectively and to dispel uncertainties, particularly regarding efficacy and side effects."

Do you think that new AI tools that can be used to screen across Social Media posts could be used to target such perception threats that can be approached in a timely manner, in a similar strategy like one used in commercial trends analysis that is used to boost managing options and sales?

Big data mining processes could uncover some perception threats or bottlenecks in adopting proactive preventive measures like immunizations in a much faster rhythm compared to conventional cross-sectional studies delivered by various platforms, like current study.

Vaccine hesitancy will probably increase globally because of polarization of communication and because of recent official statements of extremely important public figures. So, we will need a more accurate and rapid pathway of exploring VH future trends, not only simple cross-sectional approaches, and newly developing AI tools could be implemented in the foreseeable research.

Reviewer #5: Thanks a lot, dear authors for addressing my comments.

I have final suggestion that might enhance the presentation of your results.

Can you please use the same nomenclature to classify the regions of data collection to WHO regions include the African Region (AFRO), the Eastern Mediterranean Region (EMRO), the South-East Asia Region (SEARO), the Region of the Americas (AMRO), the Western Pacific Region (WPRO), and the European Region (EURO)

Reviewer #6: Overall: Accept. The author has revised the manuscript and made substantial changes based on the reviewer feedback. The document is now acceptable for publication.

Remaining Correction:

Page 13 of Results (23 of Document: “Approximately 30%” or “Exactly 30.4%” not Exact to start sentence

**Do you want your identity to be public for this peer review?** For information about this choice, including consent withdrawal, please see our Privacy Policy

Reviewer #3: **Yes: ** Mihai Craiu MD PhD

Reviewer #5: No

Reviewer #6: **Yes: ** C. Jason McKnight

---

## [Editor Report · Acceptance letter]

PONE-D-24-28390R2

PLOS ONE

Dear Dr. Ghazy,

I'm pleased to inform you that your manuscript has been deemed suitable for publication in PLOS ONE. Congratulations! Your manuscript is now being handed over to our production team.

Kind regards,

on behalf of

Dr. Stephen Dajaan Dubik

Academic Editor

PLOS ONE